# Excited Pfaffians: Generalized Neural Wave Functions Across Structure and State

**Nicholas Gao** [* 1 2]  **Till Grutschus** [* 3]  **Frank Noé** [3 4 5]  **Stephan Günnemann** [1]

## Abstract

Neural-network wave functions in Variational Monte Carlo (VMC) have achieved great success in accurately representing both ground and excited states. However, achieving sufficient numerical accuracy in state overlaps requires increasing the number of Monte Carlo samples, and consequently the computational cost, with the number of states. We present a nearly constant sample-size approach, *Multi-State Importance Sampling* (MSIS), that leverages samples from all states to estimate pairwise overlap. To efficiently evaluate all states for all samples, we introduce *Excited Pfaffians*. Inspired by Hartree-Fock, this architecture represents many states within a single neural network. Excited Pfaffians also serve as generalized wave functions, allowing a single model to represent multi-state potential energy surfaces. On the carbon dimer, we match the $\mathcal{O}(N_s^4)$-scaling natural excited states while training $> 200\times$ faster and modeling 50% more states. Our favorable scaling enables us to be the first to use neural networks to find all distinct energy levels of the beryllium atom. Finally, we demonstrate that a single wave function can represent excited states across various molecules.

## 1. Introduction

Recently, the *in-silico* design of materials and molecules has attracted significant interest for accelerating discovery (Zhang et al., 2025). Central to assessing a molecule's

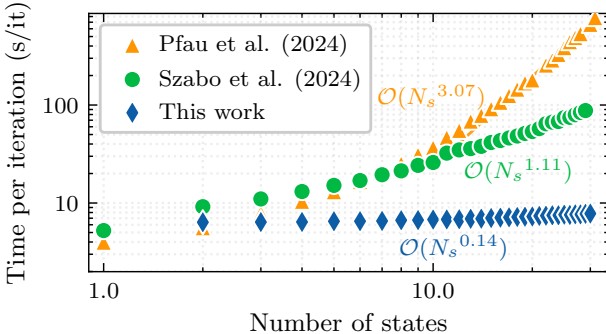

*Figure 1.* Computational scaling with the number of states. While previous works scale superlinearly, our method exhibits near-constant scaling $\mathcal{O}(N_s^{0.14})$.

properties is the electronic Schrödinger equation

$$\hat{H}\Psi_s = E_s\Psi_s \qquad (1)$$

where $\hat{H} : \mathcal{L}^2 \to \mathcal{L}^2$ is the Hamiltonian operator encoding the system's total energy and $\Psi_s : \mathbb{R}^{N_e \times 3} \to \mathbb{R}$ is the wave function describing the behavior of the system's $N_e$ electrons (Schrödinger, 1926). The eigenvalues $E_0 \leq E_1 \leq \ldots$ are the system's energy levels. While the ground state $\Psi_0$, associated with $E_0$, represents the most stable electronic configuration, excited states $\Psi_1, \Psi_2, \ldots$ govern photochemical processes, spectroscopic properties, and reaction pathways. Accurate computation of excited states is essential for applications ranging from drug design to photovoltaics. Unfortunately, analytical solutions to the Schrödinger equation exist only for the simplest systems.

Neural-network wave functions within the variational Monte Carlo (VMC) framework have achieved unprecedented accuracy for both ground and excited states (Hermann et al., 2023; Pfau et al., 2024; Hermann et al., 2020; Pfau et al., 2020). However, a single calculation may take days on contemporary hardware, and the standard approach of performing independent calculations for each system and state multiplies this cost drastically. For ground states, recent work has amortized training costs across molecular structures by learning generalized wave functions (Scherbela et al., 2024; Gao & Günnemann, 2024; Foster et al., 2025). No such scheme exists for excited states, where current methods scale superlinear to quartically with the number of

---

*Equal contribution  [1]Department of Computer Science & Munich Data Science Institute, Technical University of Munich  [2]CuspAI  [3]Free University of Berlin  [4]Rice University  [5]Microsoft Research AI4Science. Correspondence to: Nicholas Gao <nicholas@cusp.ai>, Till Grutschus <till.grutschus@fu-berlin.de>.

*Proceedings of the 43rd International Conference on Machine Learning*, Seoul, South Korea. PMLR 306, 2026. Copyright 2026 by the author(s).

states. This unfavorable scaling has two roots: (1) estimating pairwise overlaps $\langle \Psi_s | \Psi_t \rangle$ requires more samples with more states, and (2) representing many states demands either separate wave functions or prohibitively large networks.

In this work, we address both limitations. To tackle the sampling problem, we introduce *Multi-State Importance Sampling* (MSIS), which pools samples from all states to estimate pairwise overlaps, reducing the average variance of the estimator from $\mathcal{O}(N_s/N_b)$ to $\mathcal{O}(1/N_b)$ for a fixed batch size of $N_b$. To tackle the representation problem, we introduce *Excited Pfaffians*, a neural wave-function architecture inspired by Hartree-Fock theory that shares most parameters across states, differing only in a lightweight state-specific selector. Together, these contributions enable near-constant computational scaling with the number of states (Fig. 1). Moreover, Excited Pfaffians integrate seamlessly into the generalized wave function framework (Gao & Günnemann, 2023a), yielding the first method capable of jointly generalizing across molecular structures, chemical compositions, and electronic states within a single neural network.

In our experiments, we find MSIS critical for stable convergence. Without it, overlap estimates exhibit high variance, causing erratic energy optimization. This enables us to match the accuracy of NES (Pfau et al., 2024) with $40\times$ fewer samples and $200\times$ less compute while modeling 12 instead of 8 states on the carbon dimer potential energy surface (PES). Our favorable scaling enables us to be the first to compute all excited states of the beryllium atom with neural networks. For the first time, we demonstrate that a single model may approximate excited states across molecular compounds. Lastly, we find that Excited Pfaffian improves ground-state performance at state crossings compared to ground-state-only methods.

## 2. Background

We seek the $N_s$ lowest-energy solutions of the electronic Schrödinger equation, Eq. (1), with Hamiltonian

$$\hat{H} = -\frac{1}{2}\sum_i \nabla_i^2 - \sum_{i,m} \frac{Z_m}{|\boldsymbol{r}_i - \boldsymbol{R}_m|} + \sum_{i<j} \frac{1}{|\boldsymbol{r}_i - \boldsymbol{r}_j|}, \quad (2)$$

comprising kinetic energy, electron-nuclear attraction, and electron-electron repulsion for $N_e = N_\uparrow + N_\downarrow$ electrons (spin-up and spin-down) and nuclei at positions $\boldsymbol{R}_m$ with charges $Z_m$. Electronic wave functions must be antisymmetric under permutation of same-spin electrons, i.e., $\Psi(\pi(\mathbf{r})) = \text{sign}(\pi)\Psi(\mathbf{r})$ for $\pi \in S_{N_\uparrow} \times S_{N_\downarrow}$. Most methods exploit the variational principle: for any unnormalized trial wave function $\psi : \mathbb{R}^{N_e \times 3} \to \mathbb{R}$,

$$E[\psi] = \frac{\int \psi(\mathbf{r})\hat{H}\psi(\mathbf{r})\mathrm{d}\mathbf{r}}{\int \psi^2(\mathbf{r})\mathrm{d}\mathbf{r}} \geq E_0, \quad (3)$$

so minimizing $E[\psi_\theta]$ over parameters $\theta$ yields an upper bound on the ground-state energy. We denote unnormalized wave functions as $\psi$ and normalized ones as $\Psi = \psi/\mathcal{Z}$ with $\mathcal{Z} = \|\psi\|$. Excited states are obtained by minimizing energy subject to orthogonality to all lower states.

**Hartree-Fock (HF)** is an approximation, which restricts $\psi$ to a single *Slater determinant* of one-electron functions $\{\phi_p : \mathbb{R}^3 \to \mathbb{R}\}_{p=1}^{N_o}$, called *molecular orbitals (MO)*. While $N_o$ molecular orbitals are optimized, $N_e$ of these are selected for the final wave function

$$\psi_{\text{HF}}(\mathbf{r}) = \det\left[\Phi_{\text{HF}}(\mathbf{r})\Pi_0\right], \quad (4)$$

where $\Phi_{\text{HF}}(\mathbf{r})_{ip} = \phi_p(\boldsymbol{r}_i)$ is the orbital matrix function and $\Pi_0 \in \mathbb{R}^{N_o \times N_e}$ is an *orbital selector*. In the case of energy-sorted MOs, $\Pi_0$ corresponds to an identity matrix followed by a zero matrix. The determinant enforces the fermionic permutation antisymmetry. The orbitals $\phi_p$ are linear combinations of basis functions $\{\xi_\mu : \mathbb{R}^3 \to \mathbb{R}\}_{\mu=1}^{N_o}$:

$$\Phi_{\text{HF}}(\mathbf{r}) = \Xi(\mathbf{r})C \quad (5)$$

with $\Xi(\mathbf{r})_{i\mu} = \xi_\mu(\boldsymbol{r}_i)$ and variational parameters $C \in \mathbb{R}^{N_o \times N_o}$. The parameters $C$ minimize $E[\psi_{\text{HF}}]$ under the orthogonality constraints $\langle \phi_p | \phi_q \rangle = \delta_{pq}$ (Roothaan, 1951). Importantly, a set of orthogonal wave functions $\psi_{\text{HF},s}$ may be constructed by replacing $\Pi_0$ with $\Pi_s \in \mathbb{R}^{N_o \times N_e}$ such that $\det\left(\Pi_s^T \Pi_t\right) = \delta_{st}$ for all $s, t$, see App. A (Szabo & Ostlund, 2012).

**Variational Monte Carlo (VMC)** recasts Eq. (3) as

$$E[\psi_\theta] = \frac{\int \psi_\theta(\mathbf{r})\hat{H}\psi_\theta(\mathbf{r})\,\mathrm{d}\mathbf{r}}{\int \psi_\theta(\mathbf{r})^2\,\mathrm{d}\mathbf{r}} = \mathbb{E}_{\mathbf{r}\sim\rho_\theta}\left[E_{\text{loc}}(\mathbf{r})\right], \quad (6)$$

where $E_{\text{loc}}(\mathbf{r}) = \frac{(\hat{H}\psi_\theta)(\mathbf{r})}{\psi_\theta(\mathbf{r})}$ is the local energy and $\rho_\theta(\mathbf{r}) \propto \psi_\theta(\mathbf{r})^2$ is sampled via Markov chain Monte Carlo (Ceperley & Alder, 1987). Since minimization is done via gradient descent, one can optimize any differentiable unnormalized wave function with VMC. See App. B for details. Since $E_{\text{loc}}$'s variance vanishes for exact eigenstates (zero-variance principle), splitting the sample budget across structures does not degrade energy accuracy (Gao & Günnemann, 2022).

**Excited states in VMC** can be obtained via penalty-based methods (Pathak et al., 2021; Entwistle et al., 2023) that minimize the energy of wave functions $\Psi_{\theta_0}, \ldots, \Psi_{\theta_{N_s-1}}$ subject to $\langle \Psi_s | \Psi_t \rangle = \delta_{st}$ through the objective

$$\mathcal{L} = \sum_{s=0}^{N_s-1} E[\Psi_{\theta_s}] + \sum_{t\neq s} \omega_{st}|\langle \Psi_{\theta_s} | \Psi_{\theta_t} \rangle|^2, \quad (7)$$

$$\omega_{st} = \begin{cases} > E_t - E_s & \text{if } E[\Psi_{\theta_s}] < E[\Psi_{\theta_t}], \\ = 0 & \text{else.} \end{cases} \quad (8)$$

The triangular shape of $\omega$ ensures convergence to pure eigenstates rather than linear combinations that would share the same loss (Wheeler et al., 2024). Since the lower bound requires true state energies, $\omega_{st}$ is typically overestimated to avoid state collapse. See App. C for an estimator of $\omega_{st}$.

The overlap can be written as an expectation in terms of

$$
\begin{aligned}
|O_{st}| = |\langle\Psi_s|\Psi_t\rangle| &= \left|\mathbb{E}_{\mathbf{r}\sim\rho_s}\left[\frac{\Psi_t(\mathbf{r})}{\Psi_s(\mathbf{r})}\right]\right| \\
&= \sqrt{\mathbb{E}_{\mathbf{r}\sim\rho_s}\left[\frac{\psi_t(\mathbf{r})}{\psi_s(\mathbf{r})}\right]\mathbb{E}_{\mathbf{r}\sim\rho_t}\left[\frac{\psi_s(\mathbf{r})}{\psi_t(\mathbf{r})}\right]}.
\end{aligned}
\tag{9}
$$

The reformulation in the second line cancels the normalizers $\mathcal{Z}_t/\mathcal{Z}_s$. Notably, for a fixed total batch size of $N_{\mathrm{b}}$, the variance of this estimator is $\frac{N_{\mathrm{s}}(1-O_{st}^2)}{2N_{\mathrm{b}}}$, see App. D. Furthermore, the random variable $\frac{\psi_t(\mathbf{r})}{\psi_s(\mathbf{r})}$ is *unbounded* and diverges for $\psi_s(\mathbf{r}) \to 0$, leading to rare but extreme outliers.

**Neural wave functions** parameterize the trial wave function in VMC using a neural network. Typically, a permutation equivariant neural network represents *many-electron* orbitals $\phi_p^{\mathrm{NN}} : \mathbb{R}^3 \times \mathbb{R}^{N_e\times 3} \to \mathbb{R}$ that are combined in a linear combination of Slater determinants to construct an antisymmetric many-electron wave function (Hermann et al., 2020). Alternatives to the Slater determinant are antisymmetric geminal powers (Lou et al., 2023) or Pfaffians (Kim et al., 2024; Gao & Günnemann, 2024).

**Generalized wave functions** approximate the eigenstates of multiple Hamiltonians that differ in nuclear positions and charges, i.e., for a set of systems $\mathcal{M} = \{(\mathbf{R}_m, \mathbf{Z}_m)\}_{m=1}^M$, a generalized wave function $\Psi$ minimizes $\sum_{m=1}^M E[\Psi_{\theta(\mathbf{R}_m, \mathbf{Z}_m)}]$ (Gao & Günnemann, 2022).

This is commonly accomplished by having the wave functions' parameters $\theta$ be a function of the nuclear positions $\mathbf{R}$ and charges $\mathbf{Z}$, typically through a meta-network that produces system-specific parameters (Gao & Günnemann, 2023a; Scherbela et al., 2024). This enables amortized training where the cost of adding new systems grows sublinearly, as the meta-network learns transferable representations across chemical space (Foster et al., 2025). Note that the transfer to unseen structures has seen less success and is out of scope of this work (Scherbela et al., 2023).

## 3. Related Work

**Neural wave functions** achieved high accuracy for single structures (Carleo & Troyer, 2017; Hermann et al., 2020; Pfau et al., 2020; Wilson et al., 2021; 2023; Lou et al., 2023; Kim et al., 2024), with further improvements in accuracy (Gerard et al., 2022; von Glehn et al., 2023) and efficiency (Li et al., 2024a; Spencer et al., 2020; Scherbela et al., 2025), but separate training per geometry is prohibitively expensive

for PES. Generalized wave functions address this for multi-structure (Scherbela et al., 2022; Gao & Günnemann, 2022) and multi-molecule computations, using hand-crafted selectors (Gao & Günnemann, 2023a; Foster et al., 2025), or HF orbitals (Scherbela et al., 2024). The Neural Pfaffian (Gao & Günnemann, 2024) replaces the Slater determinant $\det(\Phi)$ with a Pfaffian $\mathrm{Pf}(\Phi A \Phi^T)$ (Bajdich et al., 2006; 2008), where $A \in \mathbb{R}^{N_o\times N_o}$ is skew-symmetric and $\Phi \in \mathbb{R}^{N_e\times N_o}$ need not be square, enabling a fully-learnable generalization of Slater-type wave functions. Here, we extend the Neural Pfaffian to efficiently represent excited states.

**Excited states** are one of the big challenges in quantum chemistry (Mai & González, 2020; Otis & Neuscamman, 2023). In VMC, excited states can be obtained through variance minimization (Otis et al., 2020; Zhao & Neuscamman, 2016; Pineda Flores & Neuscamman, 2019), by definition (Dash et al., 2021; Cuzzocrea et al., 2020), through symmetry (Choo et al., 2018), or via overlap penalties (Pathak et al., 2021), which were the first method applied to neural wave functions (Entwistle et al., 2023). While most require *a priori* knowledge about the target state, overlap penalties are flexible but require careful tuning (Wheeler et al., 2024). Alternatively, natural excited states (NES) (Pfau et al., 2024) avoids explicit orthogonalization but scales $O(N_{\mathrm{s}}^4)$. When spin states are known, spin penalties may be used to target specific states (Szabó et al., 2024; Li et al., 2024b). Recent work extends penalty-based methods to PES (Schätzle et al., 2025). We extend the method further to generalization across molecules and improve scaling in $N_{\mathrm{s}}$.

**Monte Carlo estimation** quality depends critically on the sampling distribution (Otis & Neuscamman, 2023). While sampling from alternative distributions has reduced variance for energy gradients (Chen & Heyl, 2024; Misery et al., 2025; Inui et al., 2021), overlap estimation remains problematic. Its sensitivity to sample size $N_{\mathrm{b}}$ degrades training (Entwistle et al., 2023), and the standard remedy is scaling samples with $N_{\mathrm{s}}$ (Entwistle et al., 2023; Szabó et al., 2024; Schätzle et al., 2025). We propose pooling samples from all states via importance sampling to stabilize overlap estimates without increasing the batch size, an approach closely related to the balance heuristic estimator in Monte Carlo rendering (Veach & Guibas, 1995; Owen & Zhou, 2000).

## 4. Method

We train a neural network $\theta : \mathbb{R}^{N_{\mathrm{n}}\times 3} \times \mathbb{N}_+^{N_{\mathrm{n}}} \times \{1, \ldots, N_{\mathrm{s}}\} \to \Omega$ that maps nuclear positions $\mathbf{R}$, charges $\mathbf{Z}$, and target state $s$ to parameters $\theta(\mathbf{R}, \mathbf{Z}, s)$ for a wave function Ansatz $\Psi_{\theta(\mathbf{R}, \mathbf{Z}, s)}$ approximating one of the $N_{\mathrm{s}}$ lowest eigenfunctions of $\hat{H}_{\mathbf{R}, \mathbf{Z}}$. While previous works (Gao & Günnemann, 2022; 2023a; Scherbela et al., 2024) demonstrated cost amortization for generalization over $\mathbf{R}$ and $\mathbf{Z}$, we demonstrate amortization for excited states.

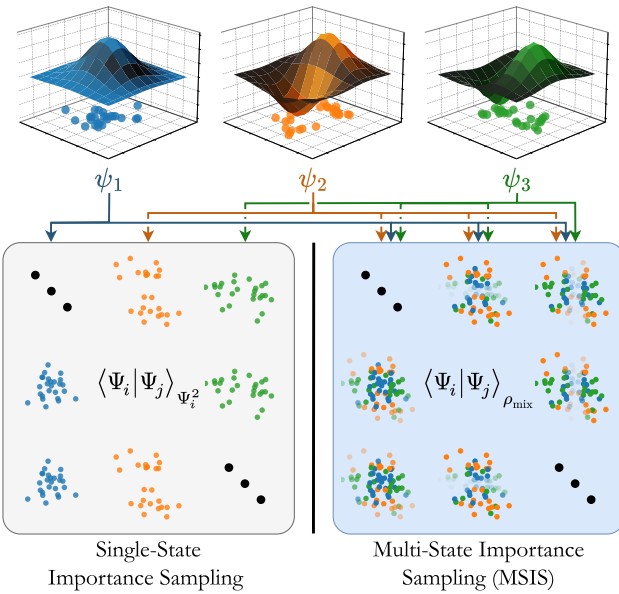

*Figure 2.* Illustration of single-state importance sampling (Szabó et al., 2024) and our Multi-State Importance Sampling (MSIS).

For favorable scaling, we keep the total number of samples $N_b$ drawn from all wave functions $\{\Psi_{\theta(\boldsymbol{R},\boldsymbol{Z},s)}\}_{\boldsymbol{R},\boldsymbol{Z},s}$ constant, so samples per wave function decrease with more structures and states. While this works for intra-wave function losses like energy (Foster et al., 2025), previous works (Entwistle et al., 2023) and our experiments (Sec. 5) show that naively applying it to the overlap degrades estimates significantly. We tackle this challenge by introducing *Multi-State Importance Sampling* (MSIS), which pools samples from all states of a structure to estimate overlaps.

However, the overlap requires evaluating all $N_s$ wave functions per structure $(\boldsymbol{R},\boldsymbol{Z})$. To accelerate this, we introduce the *Excited Pfaffian* architecture, inspired by Hartree-Fock theory, that maximizes parameter reuse across states. We additionally introduce a spin-state snapping loss to prevent convergence to linear combinations of states with different spin, and a pretraining scheme to ensure state continuity across geometric perturbations. We conclude by describing how the architecture generalizes across molecules.

**Multi-State Importance Sampling (MSIS).** The variance of the single-state importance sampling estimator in Eq. (9) scales linearly in the number of states $N_s$. Previous works scaled the total number of walkers $N_b \propto N_s$ to keep the variance constant, thereby increasing compute time.

Instead of increasing $N_b$, we pool samples from all states $\rho_{\text{mix}} = \frac{1}{N_s}\sum_{s=1}^{N_s}\rho_s$ and rewrite the inner product as

$$\langle\Psi_s|\Psi_t\rangle = \mathbb{E}_{\mathbf{r}\sim\rho_{\text{mix}}}\left[\frac{\Psi_t(\mathbf{r})\Psi_s(\mathbf{r})}{\rho_{\text{mix}}(\mathbf{r})}\right]. \quad (10)$$

As depicted in Fig. 2, this improves coverage yielding three crucial benefits. (1) The random variable $\frac{\Psi_t(\mathbf{r})\Psi_s(\mathbf{r})}{\rho_{\text{mix}}(\mathbf{r})}$ is

bounded to $\leq \frac{N_s}{2}$, eliminating the heavy-tailed outliers near wave function nodes. (2) The average overlap variance reduces from $\mathcal{O}(\frac{N_s}{N_b})$ to $\mathcal{O}(\frac{1}{N_b})$, removing the need to scale batch size with the number of states. (3) For states with vanishing *Bhattacharyya coefficient* $\langle|\Psi_s||\Psi_t|\rangle \to 0$, i.e., low unsigned overlap, the variance vanishes, whereas the single-state estimator retains unit variance. We refer to App. D for proofs and App. F for gradients.

However, since $\psi_s$ are unnormalized, we need to estimate the normalizer ratios $\kappa_s = \frac{\mathcal{Z}_1^2}{\mathcal{Z}_s^2}$ to evaluate $\frac{\Psi_t(\mathbf{r})\Psi_s(\mathbf{r})}{\rho_{\text{mix}}(\mathbf{r})}$. We accomplish this via multi-distribution bridge sampling (Meng & Wong, 1996). In short, $\boldsymbol{\kappa}$ can be estimated as the fixed point of the following system of equations:

$$B_{st}^{(\boldsymbol{\kappa})} = \begin{cases} \sum_{s'\neq s}\mathbb{E}_{\rho_{s'}}[w_s^{(\boldsymbol{\kappa})}] & \text{, if } s = t, \\ -\mathbb{E}_{\rho_s}[w_t^{(\boldsymbol{\kappa})}] & \text{, else,} \end{cases} \quad (11)$$

$$b_s^{(\boldsymbol{\kappa})} = \mathbb{E}_{\rho_s}[w_1^{(\boldsymbol{\kappa})}]. \quad (12)$$

where $q_{\text{mix}}^{(\boldsymbol{\kappa})} = \frac{1}{N_s}\sum_s \kappa_s q_s$, $w_s^{(\boldsymbol{\kappa})} = \frac{q_s}{N_s q_{\text{mix}}^{(\boldsymbol{\kappa})}}$, $q_s = \psi_s^2$. We initialize $\boldsymbol{\kappa}^{(0)} = \boldsymbol{1}$ and iterate $\hat{\boldsymbol{\kappa}}^{(t+1)} = (\boldsymbol{B}^{(t)})^{-1}\boldsymbol{b}^{(t)}$ until reaching the fixed point. For further details, we refer the reader to App. G. As the overlap, with or without MSIS, requires evaluating all wave functions for all samples, MSIS comes at no meaningful runtime cost (App. G.1).

**Excited Pfaffian.** We seek to parametrize many *orthogonal* wave functions that generalize to arbitrary molecules. We start by following the Neural Pfaffian architecture (Gao & Günnemann, 2024) that defines the wave function as a linear combination of Pfaffian wave functions

$$\psi(\mathbf{r}) = \sum_{k=1}^{N_k} \text{Pf}\left(\Phi(\mathbf{r})_k A_k \Phi(\mathbf{r})_k^T\right) \quad (13)$$

where the elements of the orbital matrices $\Phi(\mathbf{r})_{kip}$ are linear readouts of the electron's embeddings $\boldsymbol{h}(\mathbf{r})_i$ and exponentially decaying envelope functions $\chi_{kp}: \mathbb{R}^3 \to \mathbb{R}$:

$$\Phi(\mathbf{r})_{kip} = \boldsymbol{h}(\mathbf{r})_i^T \boldsymbol{w}_{kp} \cdot \chi_{kp}(\boldsymbol{r}_i). \quad (14)$$

The electron embeddings are obtained through permutation equivariant architectures like graph neural networks (Hermann et al., 2020; Gao & Günnemann, 2023a), transformers (von Glehn et al., 2023), or deep sets (Pfau et al., 2020). The skew-symmetric antisymmetrizer $A_k$, the weights $\boldsymbol{w}_{kj}$ and the envelopes $\chi_{kj}$ are constructed on a per-nucleus basis and concatenated such that their number grows with the system size, ensuring that $N_o \geq N_e$.

A trivial extension to many states is to expand $\Phi_{skp}$ and consequently $\boldsymbol{w}_{skp}, \chi_{skp}$ with an additional dimension running over the number of states $N_s$, yielding $\psi_s(\mathbf{r}) = \sum_{k=1}^{N_k} \text{Pf}\left(\Phi(\mathbf{r})_{sk} A_{sk} \Phi(\mathbf{r})_{sk}^T\right)$ (Schätzle et al., 2025; Pfau et al., 2024). While simple, this definition incurs a high

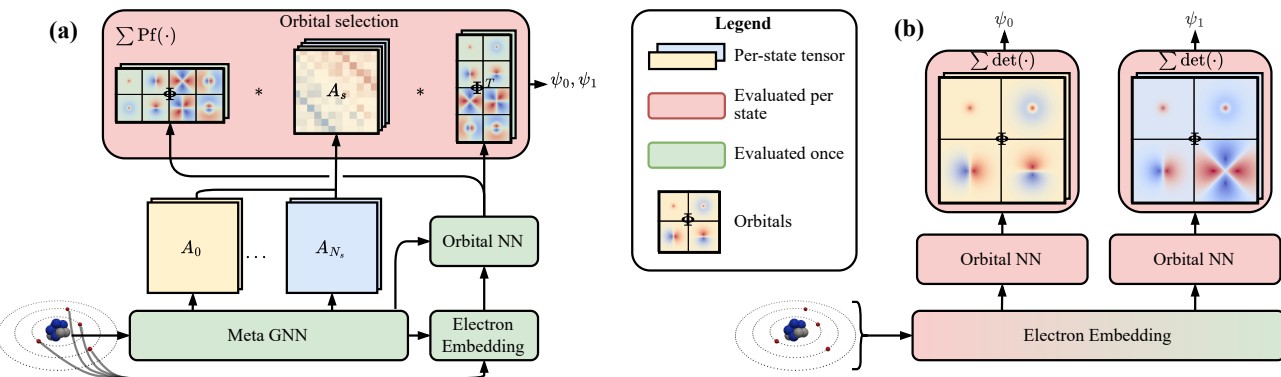

*Figure 3.* Overview of wave function architectures for excited states. (a) Our Excited Pfaffian architecture shares a single backbone network across all states, producing shared orbitals $\Phi_{\text{Pf}}$ and state-specific selector matrices $A_s$ that combine via $\text{Pf}(\Phi A_s \Phi^T)$. (b) The standard Slater determinant approach requires separate neural network passes and learned orbitals $\Phi_s$ for each state.

memory usage when applying approximate second-order methods due to the large number of parameters (Goldshlager et al., 2024). For instance, for typical choices of hyperparameters (Tab. 7), the Jacobian of the linear projection alone may require 1TB of VRAM, see App. H.

Fortunately, as we outlined in Sec. 2, wave functions do not need to differ in many parameters to be orthogonal. In HF, orthogonal wave functions can be constructed by simply changing the orbital selector $\Pi_s$. By applying the identity $\text{Pf}(BAB^T) = \det(B)\text{Pf}(A)$, one can see that we may represent these within the Neural Pfaffian framework

$$\psi_{\text{HF},s} = \det(\Phi_{\text{HF}}\Pi_s) \propto \text{Pf}(\Phi_{\text{HF}}\Pi_s A \Pi_s^T \Phi_{\text{HF}}^T) \quad (15)$$

for any non-singular skew-symmetric $A \in \mathbb{R}^{N_e \times N_e}$.

Thus, we generalize Hartree-Fock and Neural Pfaffian with our set of orthogonal wave functions

$$\psi_s(\mathbf{r}) = \sum_{k=1}^{N_k} \text{Pf}\left(\Phi(\mathbf{r})_k A_{sk} \Phi(\mathbf{r})_k^T\right) \quad (16)$$

which only differ in their orbital selection $A_{sk}$. We drop the spin-shared orbitals (Gao & Günnemann, 2023b) and use separate parameters per spin for the orbitals to allow for non-symmetric excitations. Fig. 3 depicts our architecture. As states only differ in $A_{sk}$, most of the forward pass is shared, enabling the efficient evaluation of all states. We empirically compare the architectures in App. I.

**Spin-state snapping.** The Hamiltonian commutes with the total spin operator $\hat{S}^2$, thus, eigenstates of $\hat{H}$ are also eigenstates of $\hat{S}^2$ with the eigenvalues $S(S+1)$ for integer or half-integer $S$ (Szabo & Ostlund, 2012). During initial experimentation, we encountered linear combinations of singlet and triplet states. While directly fitting the target eigenvalue is possible and resolves this (Li et al., 2024b), the spin states of the $N_s$ lowest states are generally unknown.

To resolve this without knowing the spin states *a priori*, we snap wave functions dynamically to their closest eigenvalue

$S^* = \arg\min_{S \in \frac{\mathbb{N}}{2}} |S(S+1) - \langle\Psi|\hat{S}^2|\Psi\rangle|$ of $\hat{S}^2$:

$$\mathcal{L}_{\text{snap}} = \lambda\left(\langle\Psi|\hat{S}^2|\Psi\rangle - S^*(S^*+1)\right)^2. \quad (17)$$

Note that naively computing $\nabla_\theta \mathcal{L}_{\text{snap}}$ requires differentiating the wave function twice (Szabó et al., 2024). Instead, we use the spin-raising operator $\hat{S}_+$ following Li et al. (2024b), exploiting the identity

$$\langle\Psi|\hat{S}^2|\Psi\rangle = \frac{(N_\uparrow - N_\downarrow)(N_\uparrow - N_\downarrow + 2)}{4} + \langle\hat{S}_+\Psi|\hat{S}_+\Psi\rangle. \quad (18)$$

Following Li et al. (2024b), gradients of $\langle\hat{S}_+\Psi|\hat{S}_+\Psi\rangle$ can be computed efficiently without second-order derivatives. To avoid strong biases to states similar to the pretraining targets, we initialize $\lambda$ as zero and ramp it up after 80k steps.

**Pretraining Excited Pfaffians.** Neural-network VMC typically uses HF solutions as supervised pretraining targets, as these already cover most of the total energy (Hermann et al., 2023). For single-structure excited states, one typically chooses HF excited determinants (Pfau et al., 2024; Szabó et al., 2024). Excited-state PESs introduce a novel challenge: matching excited determinants across structures to obtain smooth, orthogonal targets. We achieve this by organizing structures into a DAG and propagating canonical orbitals via basis projection, see App. J.

When matching these targets $\{\Xi, C, \Pi_s\}_{s=1}^{N_s}$ to our wave function, we need to account for disagreement in orbital ordering (Gao & Günnemann, 2023a), which we discuss in the following for a single structure and determinant. Instead of Gao & Günnemann (2024)'s alternating optimization loops, we propose a closed-form solution to pretraining generalized Pfaffian wave functions. For the orbitals $\Phi_{\text{HF}} = \Xi C$, we minimize

$$\mathcal{L}_{\text{MO}} = \|\Phi R - \Phi_{\text{HF}}\|_2^2 \quad (19)$$

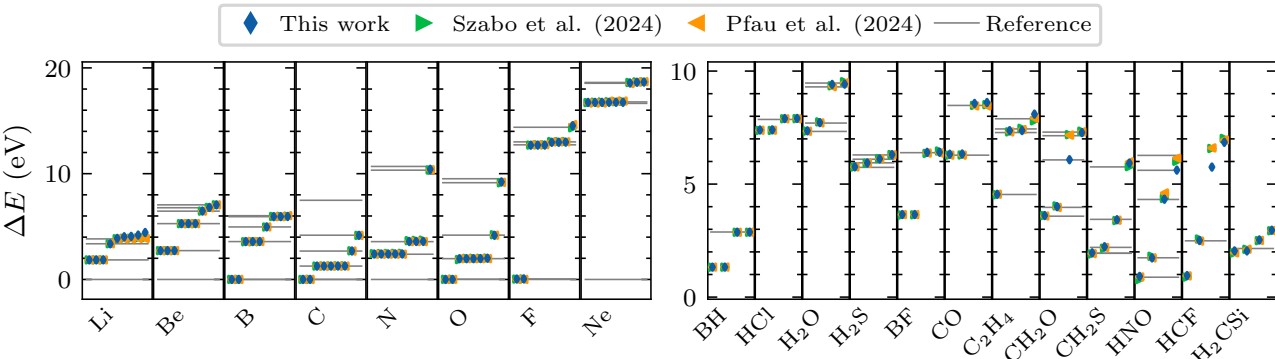

*Figure 4.* Second-row atoms' (left) and molecules' (right) excitation energies. We trained an Excited Pfaffian for each group and compare to individual models (Pfau et al., 2024; Szabó et al., 2024). References from NIST (Sansonetti, 2003) or QUEST (Véril et al., 2021).

where $R \in O(N_o)$ aligns the orbitals. This is an orthogonal Procrustes problem with solution $R^* = UV^T$ from the SVD $U\Sigma V^T = \sum_{n=1}^{N_b} \Phi(\mathbf{r}_n)^T \Phi_{HF}(\mathbf{r}_n)$. When rotating the orbitals, the orbital selector transforms to $R^*\Pi_s$, represented through $A_s$ in our Excited Pfaffians. We define

$$\mathcal{L}_A = \sum_{s=1}^{N_s} \|R\Pi_s \tilde{A}_{HF,s} \Pi_s^T R^T - A_s\|_2^2 \qquad (20)$$

where $\tilde{A}_{HF,s} \in \mathbb{R}^{N_e \times N_e}$ is an orthogonal skew-symmetric matrix representing the order ambiguity of selected orbitals. This Procrustes problem has minimizer $\tilde{A}_{HF,s}^* = \tilde{U}_s \tilde{V}_s^T$ from the SVD $\tilde{U}_s \tilde{\Sigma}_s \tilde{V}_s^T = \Pi_s^T R^T A_s R\Pi_s$. We differentiate through $R^*$ but not $\tilde{A}_{HF,s}^*$, as the repeated singular values (0 or 1) have no well-defined SVD gradient.

**Generalizing over systems.** Following the Neural Pfaffian framework (Gao & Günnemann, 2024), we generalize across molecules by conditioning a subset of wave function parameters on the nuclear configuration $(\mathbf{R}, \mathbf{Z})$. A graph neural network operating on the nuclei predicts structure-dependent parameters: nuclear embeddings, envelope decays and combinations, orbital projection weights $\mathbf{w}_{kp}$, and the antisymmetrizer $A_{sk}$. Most parameters of the electron embedding network remain shared across all systems. To ensure the number of orbitals scales with system size, we predict $N_{orb/nuc}$ orbitals per nucleus and concatenate them, yielding $N_o = N_n \cdot N_{orb/nuc}$. The antisymmetrizer $A_{sk} \in \mathbb{R}^{N_o \times N_o}$ is constructed from $N_n^2$ blocks $A_{sk}^{(mn)} \in \mathbb{R}^{N_{orb/nuc} \times N_{orb/nuc}}$, one for each pair of nuclei.

**Limitations.** While we successfully reconstruct many-state PES, our pretraining scheme does not guarantee state continuity. Depending on the training dynamics, states may still exhibit rapidly changing electronic characteristics due to small geometric perturbations, or be biased toward higher-energy states that are structurally more similar to the pretraining target. More generally, perfect state continuity is unattainable near conical intersections, where adiabatic states become degenerate, and their character changes dis-

continuously (Yarkony, 1996). Like previous works (Gerard et al., 2025; Foster et al., 2025), accuracy may degrade for distinct, uncorrelated compounds in multi-structure training, see Sec. 5. Finally, while the spin-state snapping loss resolves many linear combinations, snapping to the wrong spin state early in training may prevent convergence to lower ones; remaining linear combinations of near-degenerate states could be addressed by diagonalizing the Hamiltonian (Pfau et al., 2024).

## 5. Experiments

We evaluate Excited Pfaffian and MSIS across six aspects: (1) runtime scaling with the number of states, (2) cross-molecule accuracy on second-row atoms and molecules trained jointly, (3) now-feasible *many* state computations, (4) excited-state PES for the carbon dimer dissociation and ethylene, (5) an ablation on MSIS, and (6) the benefits in ground states when those cross other states.

Where possible, we compare our results with traditional quantum-chemical methods, experimental baselines, and previous works (Pfau et al., 2024; Szabó et al., 2024; Schätzle et al., 2025). Gradients are preconditioned using the SPRING optimizer (Goldshlager et al., 2024), and where targeting specific spin sectors was necessary, we employed the $S_+$-penalty (Li et al., 2024b). A more extensive description of the experimental setup and hyperparameters is given in App. O. Numerical results are available in App. P.

**Computational scaling.** Like Pfau et al. (2024) and Szabó et al. (2024), we measure the step time of Excited Pfaffian on a single Neon atom for 1 to 30 states. Where previous work had to measure with a batch size of 64 to fit the large number of states (Pfau et al., 2024), we measured with a batch size of 4096, as used in practice, and upscaled the runtimes from Pfau et al. (2024); Szabó et al. (2024). The results in Fig. 1 show that, in contrast to previous work, MSIS enables near-constant step-time scaling with the number of states.

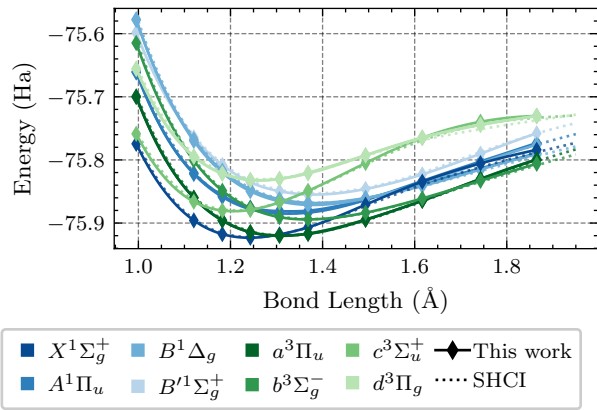

Figure 5. Potential energy surfaces for the lowest 12 states of $C_2$. Results compared against SHCI. ── singlet, ── triplet.

**Generalization across compounds.** We train an Excited Pfaffian on all second-row atoms (Li–Ne) targeting 10 states, and on 12 molecules (6–24 electrons) with 5 states each. We use a fixed batch size of $\sim$4000, amounting to $\sim$50 samples per state per atom and $\sim$70 per state per molecule, compared to 4096 (Pfau et al., 2024) and 2048 (Szabó et al., 2024) samples in prior work, respectively. To enforce pure spin eigenstates on molecules, we employed the spin-state snap loss. For elements heavier than neon, we use pseudopotentials (Li et al., 2022).

On atoms (Fig. 4, left), Excited Pfaffian captures all 10 low-lying states across all atoms. All energies, except for the 4 highest-lying states of lithium and the highest state of nitrogen and oxygen, match NIST experimental data (Sansonetti, 2003) up to chemical accuracy ($< 1.6$ mHa) and prior VMC results (Pfau et al., 2024; Szabó et al., 2024). Compared to Szabó et al. (2024) and NES requiring separate optimizations per atom and spin sector (16B/64B total samples), our joint training uses 800M samples, a $\sim$20-80$\times$ reduction. In terms of accuracy, we match the theoretical best estimates from QUEST (Véril et al., 2021) for most molecules. While fewer, we still observe Excited Pfaffian converging to linear combinations of eigenstates on $H_2O$, $H_2CSi$, and $C_2H_4$. Notably, on $CH_2O$ and HNO we find states missed by previous methods. Joint training reduces samples from 12B and 48B to 800M ($\sim$15-60$\times$) compared to Szabó et al. (2024) and NES, respectively.

**Many excitations.** Due to their unfavorable scaling, previous works on neural network-based excited states were limited to $<$ 10 excited states. Our favorable scaling changes this and enables an unprecedented number of excited states. We demonstrate this on the beryllium atom by computing *all* 32 excited states up to the ioniza-

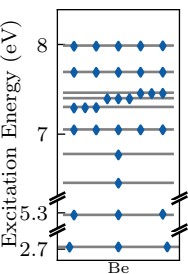

Figure 6. 32 excited states of Be.

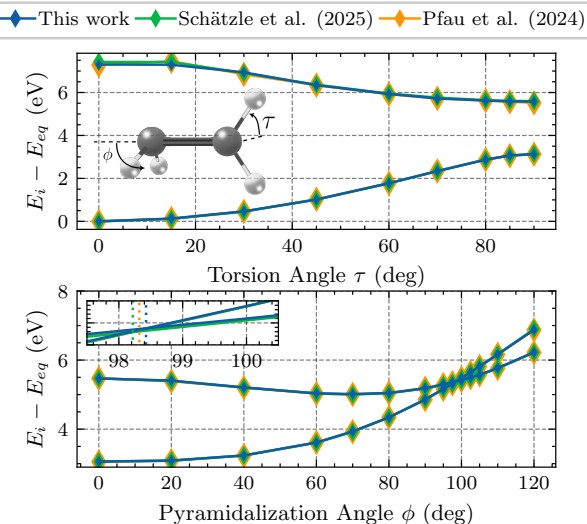

Figure 7. Potential energy surfaces of ethylene along torsion ($\tau$) and pyramidalization ($\phi$). Top: rotation about the C–C bond. Bottom: pyramidalization showing a state crossing $\phi \approx 99°$ (inset).

tion potential. While higher states exist mathematically, beyond this threshold, a real electron would be ejected from the atom. In Fig. 6, Excited Pfaffian recovers all 33 states from the NIST database (Sansonetti, 2003) with errors $< 0.5$ mHa.

**Excited-state potential energy surfaces.** The carbon dimer is a challenging benchmark due to strong correlation and numerous low-lying state crossings along its dissociation curve. We compute the $C_2$ PES from $0.995$ Å to $1.87$ Å across 10 structures from Pfau et al. (2024), jointly training a single Excited Pfaffian on 6 singlet and 6 triplet states enforced via the $S_+$ penalty. This amounts to only $\sim$32 samples per state and structure, yet we model 12 states compared to the 8 covered by previous neural VMC works (Pfau et al., 2024; Schätzle et al., 2025). In total, we use 800M samples during optimization, compared to 32B for NES (Pfau et al., 2024) and 3.2B for Schätzle et al. (2025).

Fig. 5 compares our results against highly accurate stochastic heat-bath configuration interaction (SHCI) calculations (Holmes et al., 2017), with energies relabeled to match the SHCI state ordering. Our accuracy is on par with the significantly more expensive NES method (Pfau et al., 2024), while Schätzle et al. (2025) deviates from the baseline at stretched geometries. Like Schätzle et al. (2025), we observe a single state swap between the structurally similar $X^1\Sigma_g^+$ and $B'^1\Sigma_g^+$ states near $1.6$ Å, which share the same symmetry and lie close in energy. This state swap reflects a narrow avoided crossing (von Neumann & Wigner, 1929) at which the dominant electronic character of the two states exchanges over a small range of bond lengths. The two higher-lying singlet and triplet states not computed in previous works also agree well with the SHCI baseline, demonstrating the extension to states beyond those previously studied.

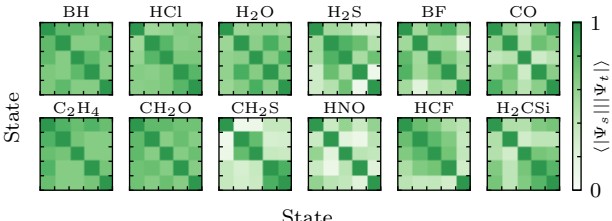

*Figure 8.* Spatial overlap measured through Bhattacharyya coefficients $\langle|\Psi_s|||\Psi_t|\rangle$ for all molecules in Fig. 4.

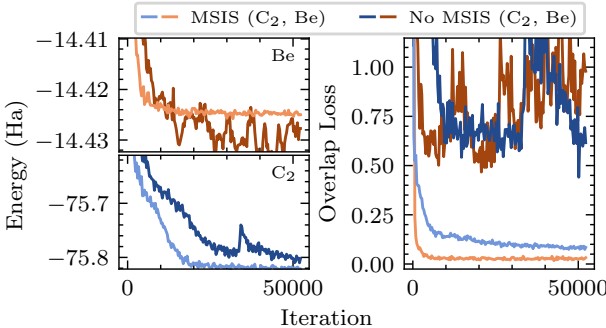

*Figure 10.* Ablation study on MSIS for the 12-state $C_2$ PES and the 33-state Be. The x-axis shows the number of VMC steps and the y-axis shows the total energy (left) and overlap (right).

App. K provides detailed comparisons to previous works.

Further, we evaluate Excited Pfaffian on ethylene along two coordinates: torsion about the C–C bond ($\tau$) and pyramidalization ($\phi$). We compute the 2 lowest singlet states, enforcing spin via the $S_+$ penalty, jointly across both panels. We observe a loss of state tracking in the torsion PES between $15°$ and $30°$, where a Rydberg state crosses the valence excited state resulting in a change in the electronic character of the upper state, see App. L (Barbatti et al., 2004). Fig. 7 shows all methods are in good agreement along both coordinates. Along pyramidalization, we locate the intersection at $\phi \approx 98.3°$, matching NES.

**MSIS ablation.** We investigate how MSIS performs across different Bhattacharyya coefficients $BC_{st}$ regimes, where $BC_{st} = \langle|\Psi_s|||\Psi_t|\rangle$ measures the similarity between $\rho_s$ and $\rho_t$. Fig. 8 plots the coefficients for all molecules in Fig. 4, revealing a wide range of values across state pairs. MSIS performs well throughout this range due to two complementary mechanisms. For pairs with high similarity, samples from one state remain informative for the other, effectively increasing the sample size for overlap estimation. For distant pairs, our estimator's variance vanishes, see App. D. We empirically confirm this variance reduction across all systems in App. E. We ablate MSIS on the 12-state $C_2$ PES and 33-state Be. Without MSIS, Fig. 10 shows that overlap estimates exhibit high variance throughout training, leading to oscillating loss and destabilizing energy optimization. With MSIS, the overlap loss drops rapidly within 5k iterations and remains stable thereafter, translating to consistent energy convergence.

**Ground-state accuracy through state crossings.** We demonstrate that Excited Pfaffian can improve ground-state accuracy when crossings occur. We train on 10 equidistant $C_2$ geometries between $1.2\,\text{Å}$ and

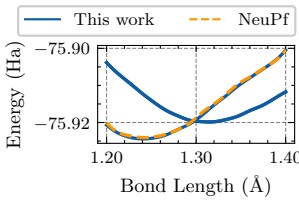

*Figure 9.* Potential energy surface of $C_2$ near a state crossing.

$1.4\,\text{Å}$, targeting 3 states to account for the $a^3\Pi_u$ state being degenerate. For comparison, we train a ground-state Neural Pfaffian (Gao & Günnemann, 2024) on the same geometries. Fig. 9 shows that the Neural Pfaffian, being a smooth function of nuclear coordinates, cannot discontinuously switch character at the $1.3\,\text{Å}$ crossing. Once converged to the $X^1\Sigma_g^+$ state on one side, it remains trapped. At an eigenstate, the energy gradient vanishes, providing no signal to descend to the true ground state. In contrast, Excited Pfaffian correctly tracks individual electronic states through the crossing, yielding the true ground-state energy at all geometries by taking the minimum across all states.

**Compute time.** So far, our discussion has focused on sample counts rather than compute time. For Szabó et al. (2024) and Schätzle et al. (2025), compute cost is proportional to the number of samples, but NES scales between $\mathcal{O}(N_s^2)$ and $\mathcal{O}(N_s^4)$. Direct comparisons are complicated by different choices of second-order optimizer (Martens, 2010; Goldshlager et al., 2024) and implementation. Direct wall-clock measurements at equal batch size (App. N) show that our method achieves $> 30\times$ and $> 200\times$ per-step speedups over NES on molecules (Fig. 4) and the carbon dimer (Fig. 5), respectively, when summed over per-structure baseline training. For the 33 states of beryllium, we require $> 300\times$ less compute than NES. At our $C_2$ compute budget, NES is far from convergence (App. N.1).

## 6. Discussion

We addressed the superlinear time-scaling of excited states in neural-network variational Monte Carlo via two main contributions. Multi-State Importance Sampling reduces the variances of overlap estimates by pooling samples across states, and the Excited Pfaffian architecture efficiently encodes many states in a single model. Jointly, these result in sublinear, near-constant time scaling in the number of excited states. Moreover, we made independently valuable contributions to neural wave functions. Spin-state snapping provides the benefits of a spin penalty without requiring prior knowledge of the correct spin sectors, and our improved pretraining procedure eases the initialization of generalized wave functions.

Taken together, these contributions enabled us to match the accuracy of up to $200\times$ more expensive methods. We computed many-state potential energy surfaces with high accuracy at a fraction of the cost, and were the first to identify all excitations of the beryllium atom using neural networks in VMC. We also demonstrated the first neural wave function that generalizes across molecules for excited states. However, while strong synergies exist in modeling energy surfaces, like previous works (Gerard et al., 2025; Gao & Günnemann, 2024), lossless generalization across chemical compounds remains an open challenge.

Nonetheless, the advances made may enable new applications of neural-network VMC in photochemical simulations (Tang et al., 2025) or in the modeling of absorption spectra (Liu et al., 2025). Beyond direct applications, generalized excited-state wave functions enable the generation of accurate reference data for training excited-state machine-learning force fields (Huang et al., 2025), or density functionals (Cheng et al., 2025; Luise et al., 2025).

## Impact Statement

Our work reduces the computational cost of simulating excited states. We do not foresee negative societal consequences beyond those common to computational chemistry.

## Acknowledgments

TG acknowledges financial support from Deutsche Forschungsgemeinschaft (DFG) through grant CRC 1114 "Scaling Cascades in Complex Systems", Project Number 235221301, Project B08 "Multiscale Boltzmann Generators". Further, the authors gratefully acknowledge the scientific support and resources of the AI service infrastructure LRZ AI Systems, provided by the Leibniz Supercomputing Centre (LRZ) of the Bavarian Academy of Sciences and Humanities (BAdW), and funded by the Bayerisches Staatsministerium für Wissenschaft und Kunst (StMWK). This research is in part funded by the Federal Ministry of Education and Research (BMBF) and the Free State of Bavaria under the Excellence Strategy of the Federal Government and the Länder. We thank Zeno Schätzle for fruitful discussions and for providing reference data.

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

## A. Hartree-Fock excited-state construction

In Hartree-Fock theory, the ground state occupies the $N_e$ lowest-energy orbitals, selected by $\Pi_0 = [I_{N_e}, \mathbf{0}]^T \in \{0,1\}^{N_o \times N_e}$. Excited states are constructed by replacing occupied orbitals with virtual (unoccupied) orbitals. For a single excitation from orbital $p$ to orbital $a > N_e$, the selector $\Pi_{p \to a}$ is obtained by swapping the $p$th and $a$th rows of $\Pi_0$.

More generally, let $\Pi_s, \Pi_t \in \{0,1\}^{N_o \times N_e}$ be two orbital selectors where each column contains exactly one 1. The resulting wave functions are orthogonal if and only if

$$\langle \Psi_{\mathrm{HF},s} | \Psi_{\mathrm{HF},t} \rangle = \det\left(\Pi_s^T \Pi_t\right) = \delta_{st}. \tag{21}$$

This follows from the fact that for Slater determinants $\Psi_s = \det(\Phi \Pi_s)$ and $\Psi_t = \det(\Phi \Pi_t)$ with orthonormal orbitals $\Phi^T \Phi = I$:

$$\langle \Psi_s | \Psi_t \rangle = \det\left(\Pi_s^T \Phi^T \Phi \Pi_t\right) = \det\left(\Pi_s^T \Pi_t\right). \tag{22}$$

When $\Pi_s$ and $\Pi_t$ select disjoint sets of orbitals in at least one position, $\Pi_s^T \Pi_t$ has a zero row, yielding $\det\left(\Pi_s^T \Pi_t\right) = 0$. When they select identical orbitals, $\Pi_s^T \Pi_t$ is a permutation matrix with $\det\left(\Pi_s^T \Pi_t\right) = \pm 1$.

More generally, this result extends to real-valued selectors $\Pi_s, \Pi_t \in \mathbb{R}^{N_o \times N_e}$: the condition $\det\left(\Pi_s^T \Pi_t\right) = \delta_{st}$ remains necessary and sufficient for orthogonality of the resulting wave functions.

## B. Variational Monte Carlo

Variational Monte Carlo optimizes a parametrized wave function $\psi_\theta$ by minimizing the energy expectation

$$E[\psi_\theta] = \mathbb{E}_{\mathbf{r} \sim \rho_\theta} \left[E_{\mathrm{loc}}(\mathbf{r})\right], \tag{23}$$

where $\rho_\theta(\mathbf{r}) = \frac{\psi_\theta(\mathbf{r})^2}{\int \psi_\theta(\mathbf{r}')^2 d\mathbf{r}'}$ and $E_{\mathrm{loc}}(\mathbf{r}) = \frac{(\hat{H}\psi_\theta)(\mathbf{r})}{\psi_\theta(\mathbf{r})}$.

**Sampling.** Samples from $\rho_\theta$ are obtained via Markov Chain Monte Carlo (MCMC), typically using the Metropolis-Hastings algorithm. Starting from a configuration $\mathbf{r}$, a proposal $\mathbf{r}'$ is drawn from a proposal distribution $q(\mathbf{r}'|\mathbf{r})$ and accepted with probability

$$A(\mathbf{r} \to \mathbf{r}') = \min\left(1, \frac{\rho_\theta(\mathbf{r}')q(\mathbf{r}|\mathbf{r}')}{\rho_\theta(\mathbf{r})q(\mathbf{r}'|\mathbf{r})}\right) = \min\left(1, \frac{\Psi_\theta(\mathbf{r}')^2 q(\mathbf{r}|\mathbf{r}')}{\Psi_\theta(\mathbf{r})^2 q(\mathbf{r}'|\mathbf{r})}\right). \tag{24}$$

For symmetric proposals $q(\mathbf{r}'|\mathbf{r}) = q(\mathbf{r}|\mathbf{r}')$, this simplifies to accepting based on the ratio $\psi_\theta(\mathbf{r}')^2/\psi_\theta(\mathbf{r})^2$.

**Energy gradients.** The gradient of the energy with respect to parameters $\theta$ is given by

$$\nabla_\theta E[\psi_\theta] = 2\mathbb{E}_{\mathbf{r} \sim \rho_\theta} \left[\left(E_{\mathrm{loc}}(\mathbf{r}) - E[\psi_\theta]\right) \nabla_\theta \log |\psi_\theta(\mathbf{r})|\right], \tag{25}$$

where the baseline $E[\psi_\theta]$ reduces variance without introducing bias. This expression is estimated using $N_b$ samples $\{\mathbf{r}_i\}_{i=1}^{N_b}$ from the MCMC chain:

$$\nabla_\theta E[\psi_\theta] \approx \frac{2}{N_b} \sum_{i=1}^{N_b} \left(E_{\mathrm{loc}}(\mathbf{r}_i) - \bar{E}\right) \nabla_\theta \log |\psi_\theta(\mathbf{r}_i)|, \tag{26}$$

where $\bar{E} = \frac{1}{N_b} \sum_i E_{\mathrm{loc}}(\mathbf{r}_i)$.

**Local energy computation.** The local energy requires evaluating $\hat{H}\psi_\theta/\psi_\theta$. For the electronic Hamiltonian, this involves:

$$E_{\mathrm{loc}}(\mathbf{r}) = -\frac{1}{2} \sum_i \frac{\nabla_i^2 \psi_\theta(\mathbf{r})}{\psi_\theta(\mathbf{r})} + V(\mathbf{r}), \tag{27}$$

where $V(\mathbf{r})$ contains the Coulomb potential terms. The kinetic term is computed as

$$\frac{\nabla_i^2 \psi_\theta}{\psi_\theta} = \nabla_i^2 \log |\psi_\theta| + \left(\nabla_i \log |\psi_\theta|\right)^2. \tag{28}$$

## C. Overlap penalty scale estimation

As shown by Pathak et al. (2021) and refined by Wheeler et al. (2024), the penalty coefficients must satisfy

$$\omega_{st} > |E_t - E_s| \tag{29}$$

to avoid collapse of the optimized states onto each other. While any value satisfying this constraint yields the correct global minimum, values closer to the bound can reduce training noise.

Following Szabó et al. (2024), we employ automatic scaling of $\omega_{st}$ based on running estimates of the energies:

$$\omega_{st} = \tilde{\omega} \cdot \max \left( |\bar{E}_s - \bar{E}_t|, \sigma(E_s), \epsilon \right), \tag{30}$$

where $\tilde{\omega} > 1$ is a shared hyperparameter, $\bar{E}_s$ denotes the exponentially weighted mean of the local energies for state $s$, and $\sigma(E_s)$ is the standard deviation. The first term ensures the constraint is satisfied while adapting to the system-specific energy gaps. The second term prevents state collapse during early training when energy estimates are unreliable. The floor $\epsilon = 10^{-3} E_h$ provides numerical stability.

This parametrization significantly reduces system-dependence compared to manually tuning individual $\omega_{st}$ values. Like Szabó et al. (2024), we use $\tilde{\omega} = 4$ for all systems in this work. For PES calculations, Schätzle et al. (2025) propose dynamical state reordering during training to maintain state continuity across geometries. We adopt the same approach. States are reordered by energy at each optimization step, ensuring that the penalty coefficients $\omega_{st}$ consistently penalize high-energy states rather than states with fixed indices.

## D. Variance of the overlap estimator

We analyze Monte Carlo estimators for the normalized overlap integral $O_{st} = \langle \Psi_s | \Psi_t \rangle$, where $\mathcal{Z}_s = \sqrt{\int \psi_s(x)^2 \mathrm{d}x}$ is the normalization constant, $\Psi_s(x) = \frac{\psi_s(x)}{\mathcal{Z}_s}$ is the normalized wave function and $\rho_s(x) = \Psi_s(x)^2$ is the corresponding probability density. Estimator 2 requires the normalizing constant ratios $\kappa_s = \mathcal{Z}_1^2 / \mathcal{Z}_s^2$; these are estimated via bridge sampling as described in G. The variance analysis below is conditional on these ratios.

### D.1. Estimator 1: Single-state distribution.

Sampling from $\rho_s$ alone yields

$$O_{st} = \mathbb{E}_{\rho_s} \left[ \frac{\psi_t(x)}{\psi_s(x)} \right] \frac{\mathcal{Z}_s}{\mathcal{Z}_t}. \tag{31}$$

The integrand $\psi_t(x)/\psi_s(x)$ is *unbounded*: wherever $\psi_s(x) \to 0$ while $\psi_t(x) \neq 0$, the ratio diverges. For orthogonal states, this occurs in extended regions where the nodal structures differ, leading to high-variance estimates.

Denote the random variable $g(x) = \psi_t(x)/\psi_s(x)$. The second moment is

$$\mathbb{E}_{\rho_s}[g^2] = \int \frac{\psi_s^2}{\mathcal{Z}_s^2} \cdot \frac{\psi_t^2}{\psi_s^2} \mathrm{d}x = \int \frac{\psi_t^2}{\mathcal{Z}_s^2} \mathrm{d}x = \frac{\mathcal{Z}_t^2}{\mathcal{Z}_s^2}, \tag{32}$$

giving variance

$$\mathrm{Var}_{\rho_s}[g] = \frac{\mathcal{Z}_t^2}{\mathcal{Z}_s^2} - \frac{\mathcal{Z}_t^2}{\mathcal{Z}_s^2} O_{st}^2 = \frac{\mathcal{Z}_t^2}{\mathcal{Z}_s^2} \left( 1 - O_{st}^2 \right). \tag{33}$$

Although this expression appears bounded, the random variable $g(x) = \psi_t(x)/\psi_s(x)$ itself is *unbounded*: it diverges wherever $\psi_s(x) \to 0$ while $\psi_t(x) \neq 0$. The finite variance arises only because such regions have measure zero under $\rho_s$. In practice, finite-sample estimates suffer from rare but extreme values when samples land near nodes of $\psi_s$, causing heavy tails and unreliable convergence.

A further complication is that this estimator requires knowledge of the normalization ratio $\frac{\mathcal{Z}_s}{\mathcal{Z}_t}$. This can be circumvented by sampling from both $\rho_s$ and $\rho_t$ and using the geometric mean. Define $A = \mathbb{E}_{\rho_s}[g]$ and $B = \mathbb{E}_{\rho_t}[g^{-1}]$. Then $AB = O_{st}^2$, so

$|O_{st}| = \sqrt{AB}$ with the normalization ratios canceling. Using the delta method with $\frac{N_{\rm b}}{N_{\rm s}}$ samples from each distribution, the per-pair variance is

$$\mathrm{Var}[\hat{O}_{st}] \approx \frac{N_{\rm s}(1 - O_{st}^2)}{2N_{\rm b}}. \tag{34}$$

This depends only on $O_{st}$, not the normalization ratio. However, the underlying random variables remain unbounded, and the delta method requires $|O_{st}|$ bounded away from zero, so this approximation may be unreliable for nearly orthogonal states.

With sample reuse across all $\binom{N_{\rm s}}{2}$ pairs, the total variance is

$$\sum_{s<t} \mathrm{Var}[\hat{O}_{st}] \approx \frac{N_{\rm s}}{2N_{\rm b}} \sum_{s<t} (1 - O_{st}^2). \tag{35}$$

For orthogonal states where $O_{st} \approx 0$, this becomes $\frac{N_{\rm s}^2(N_{\rm s}-1)}{4N_{\rm b}}$, scaling as $\mathcal{O}\big(\frac{N_{\rm s}^3}{N_{\rm b}}\big)$.

### D.2. Estimator 2: $N_{\rm s}$-state mixture.

When optimizing $N_{\rm s}$ states simultaneously, we estimate all $\binom{N_{\rm s}}{2}$ pairwise overlaps from a single $N_{\rm s}$-component mixture $\rho_{\rm mix}(x) = \frac{1}{N_{\rm s}} \sum_{s=1}^{N_{\rm s}} \Psi_s(x)^2$. The pairwise overlap becomes

$$O_{st} = \mathbb{E}_{\rho_{\rm mix}}[f_{st}(x)], \quad \text{where} \quad f_{st}(x) = \frac{N_{\rm s} \cdot \Psi_s(x)\Psi_t(x)}{\sum_{s'=1}^{N_{\rm s}} \Psi_{s'}(x)^2}. \tag{36}$$

By AM-GM, $\sum_{s'} \Psi_{s'}^2 \geq \Psi_s^2 + \Psi_t^2 \geq 2|\Psi_s \Psi_t|$, so $|f_{st}(x)| \leq \frac{N_{\rm s}}{2}$ is *bounded*.

**Per-pair variance bound.** The second moment of $f_{st}$ is

$$\mathbb{E}_{\rho_{\rm mix}}[f_{st}^2] = \int \frac{1}{N_{\rm s}} \sum_u \Psi_u^2 \cdot \frac{N_{\rm s}^2 \Psi_s^2 \Psi_t^2}{(\sum_{s'} \Psi_{s'}^2)^2} \mathrm{d}x = N_{\rm s} \int \frac{\Psi_s^2 \Psi_t^2}{\sum_{s'} \Psi_{s'}^2} \mathrm{d}x. \tag{37}$$

Using the AM-GM bound $\sum_{s'} \Psi_{s'}^2 \geq 2|\Psi_s||\Psi_t|$:

$$\mathbb{E}_{\rho_{\rm mix}}[f_{st}^2] \leq N_{\rm s} \int \frac{|\Psi_s|^2|\Psi_t|^2}{2|\Psi_s||\Psi_t|} \mathrm{d}x = \frac{N_{\rm s}}{2} \int |\Psi_s||\Psi_t| \mathrm{d}x = \frac{N_{\rm s}}{2} \mathrm{BC}_{st}, \tag{38}$$

where $\mathrm{BC}_{st} = \langle |\Psi_s| | |\Psi_t| \rangle$ is the Bhattacharyya coefficient (overlap of unsigned wave functions). The per-pair variance with $N_{\rm b}$ samples is therefore bounded by

$$\mathrm{Var}[\hat{O}_{st}] = \frac{\mathbb{E}[f_{st}^2] - O_{st}^2}{N_{\rm b}} \leq \frac{N_{\rm s} \mathrm{BC}_{st} - 2O_{st}^2}{2N_{\rm b}}. \tag{39}$$

For orthogonal states ($O_{st} = 0$), this simplifies to $\mathrm{Var}[\hat{O}_{st}] \leq \frac{N_{\rm s}\mathrm{BC}_{st}}{2N_{\rm b}}$. States with low coefficients (spatially disjoint) have lower variance, states with high coefficients contribute more. Since $O_{st} \leq \mathrm{BC}_{st} \leq 1$, the per-pair bound scales as $\mathcal{O}(N_{\rm s}/N_{\rm b})$. However, analyzing all pairs jointly yields a tighter result.

**Joint variance bound.** The sum of second moments is

$$\sum_{s<t} \mathbb{E}_{\rho_{\rm mix}}[f_{st}^2] = N_{\rm s} \sum_{s<t} \int \frac{\Psi_s^2 \Psi_t^2}{\sum_{s'} \Psi_{s'}^2} \mathrm{d}x. \tag{40}$$

Using $(\sum_{s'} \Psi_{s'}^2)^2 = \sum_{s'} \Psi_{s'}^4 + 2\sum_{s<t} \Psi_s^2 \Psi_t^2$, we obtain

$$\sum_{s<t} \mathbb{E}_{\rho_{\rm mix}}[f_{st}^2] = \frac{N_{\rm s}}{2} \int \left( \sum_{s'} \Psi_{s'}^2 - \frac{\sum_{s'} \Psi_{s'}^4}{\sum_{s'} \Psi_{s'}^2} \right) \mathrm{d}x = \frac{N_{\rm s}}{2}(N_{\rm s} - P), \tag{41}$$

where $P := \int \frac{\sum_{s'} \Psi_{s'}^4}{\sum_{s'} \Psi_{s'}^2} dx$. $P$ satisfies $1 \leq P \leq N_s$: the lower bound follows from Cauchy-Schwarz $(\sum_{s'} (\Psi_{s'}^2)^2 \cdot N_s \geq (\sum_{s'} \Psi_{s'}^2)^2$ applied pointwise), and the upper bound from $\Psi_{s'}^4 \leq \Psi_{s'}^2 \sum_{t'} \Psi_{t'}^2$. Dividing by $\binom{N_s}{2}$ pairs gives the average second moment:

$$\frac{1}{\binom{N_s}{2}} \sum_{s<t} \mathbb{E}_{\rho_{mix}}[f_{st}^2] = \frac{N_s - P}{N_s - 1} \leq 1. \tag{42}$$

With $N_b$ total samples ($\frac{N_b}{N_s}$ per state), the sum of Monte Carlo variances satisfies

$$\sum_{s<t} \text{Var}[\hat{O}_{st}] = \frac{\sum_{s<t}(\mathbb{E}[f_{st}^2] - O_{st}^2)}{N_b} = \frac{N_s(N_s - P)}{2N_b} - \frac{\sum_{s<t} O_{st}^2}{N_b} \leq \frac{N_s(N_s - 1)}{2N_b}. \tag{43}$$

For orthogonal states, this scales as $\mathcal{O}\left(\frac{N_s^2}{N_b}\right)$, a factor of $\frac{N_s}{2}$ improvement over Estimator 1.

**Effect of estimated normalization ratios.**  The analysis above conditions on the true ratios $\kappa_s = \frac{\mathcal{Z}_1^2}{\mathcal{Z}_s^2}$. In practice, these are estimated via bridge sampling (App. G), introducing additional variance. For two states, bridge sampling quality degrades when the unsigned overlap $\text{BC}_{st}$ decreases, as the distributions have less overlapping support. However, with many states, a state only needs spatial overlap with *any* of the others, not all. Even if $\text{BC}_{st}$ is small, intermediate states can bridge the gap.

## E. Effective sample size of MSIS

The MSIS effective sample size depends on the Bhattacharyya coefficient $\text{BC}_{st} = \langle|\Psi_s|||\Psi_t|\rangle$ between states. At $\text{BC}_{st} = 0$, the wave functions have disjoint support and samples from $\Psi_s$ do not contribute to overlap estimates involving $\Psi_t$. At $\text{BC}_{st} = 1$, the two distributions coincide and every sample contributes fully to both states. We investigate how this effective sample size scales with system size. For each state $s$ we use the Kish effective sample size (Kish, 1965) of the MSIS reweighting factor $w_s(\mathbf{r}) = q_s \kappa_s / q_{mix}$:

$$\text{ESS}_s = \frac{\left(\sum_{n=1}^{N_b} w_s(\mathbf{r}_n)\right)^2}{\sum_{n=1}^{N_b} w_s(\mathbf{r}_n)^2}, \quad \mathbf{r}_n \sim \rho_{mix}, \tag{44}$$

which counts how many of the $N_b$ mixture samples effectively contribute to the MSIS estimate of $O_{st}$. We report the normalized quantity $\text{ESS} \cdot N_s / N_b$, chosen so that single-state importance sampling, which effectively yields $N_b/N_s$ samples per pair, corresponds to 1, while maximal pooling, where every sample contributes equally to every pair, corresponds to $N_s$. Fig. 11 reports this quantity for the second-row atoms and the 12 molecules of Fig. 4. For atoms ($N_s = 10$), the normalized ESS ranges from $\sim 2.3$ to $\sim 4.7$, and for molecules ($N_s = 5$) from $\sim 1.8$ to $\sim 2.9$. All systems remain well above the single-state baseline, empirically confirming the variance reduction of MSIS. Crucially, the observed variation reflects differences in electronic character rather than system size. Normalized ESS is flat with the number of atoms (right panel) and with valence electrons within each system class (middle panel). We leave the systematic study at larger system sizes for future work.

## F. Gradients of the overlap estimator

The gradient of the pairwise overlap with respect to the network parameters $\theta$ can be written as (Entwistle et al., 2023)

$$\nabla_\theta O_{st} = \mathbb{E}_{\rho_s}\left[\left(\frac{\psi_t(\mathbf{r})}{\psi_s(\mathbf{r})} - \mathbb{E}_{\rho_s}\left[\frac{\psi_t(\mathbf{r})}{\psi_s(\mathbf{r})}\right]\right) \nabla_\theta \ln \psi_s(\mathbf{r})\right] \mathbb{E}_{\rho_t}\left[\frac{\psi_s(\mathbf{r})}{\psi_t(\mathbf{r})}\right]. \tag{45}$$

This expression requires accurate estimates of the overlap ratios $\mathbb{E}_{\rho_s}[\frac{\psi_t}{\psi_s}]$ and $\mathbb{E}_{\rho_t}[\frac{\psi_s}{\psi_t}]$. When estimated from single-state samples, these ratios exhibit high variance early in training, leading to noisy gradients that destabilize optimization (see Sec. 5). We address this by computing the expectation over the same pooled distribution $\rho_{mix}$ used for the overlap itself, replacing the single-state overlap estimates with our MSIS estimator from Eq. (10):

$$\nabla_\theta O_{st} = \mathbb{E}_{\rho_s}\left[\left(\frac{\psi_t(\mathbf{r})}{\psi_s(\mathbf{r})} - \hat{O}_{st}\right) \nabla_\theta \ln \psi_s(\mathbf{r})\right] \hat{O}_{ts}, \tag{46}$$

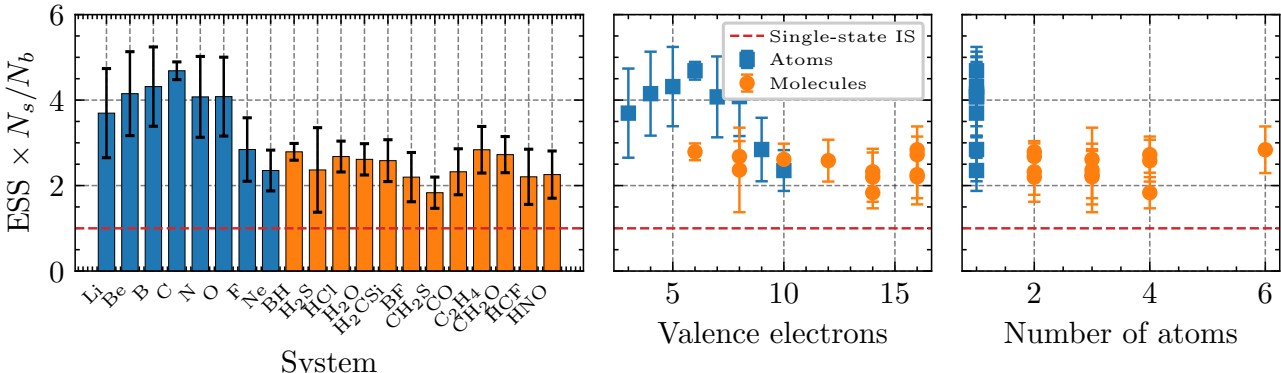

*Figure 11.* Normalized effective sample size ESS $\cdot$ $N_s/N_b$ of the MSIS overlap estimator across the second-row atoms (blue, $N_s = 10$) and 12 molecules (orange, $N_s = 5$). Left: per system, averaged over states with error bars indicating the standard deviation. Middle: as a function of the number of valence electrons. Right: as a function of the number of atoms. The dashed red line at 1 marks the effective sample size of single-state importance sampling.

where $\hat{O}_{st} = \mathbb{E}_{\rho_{\text{mix}}}\left[\frac{\psi_t(\mathbf{r})\psi_s(\mathbf{r})\kappa_s}{q_{\text{mix}}}\right]$ is the MSIS estimate of the single-state ratio $\mathbb{E}_{\rho_s}[\psi_t/\psi_s]$. We also experimented with using $\rho_{\text{mix}}$ for the outer expectation, but found it made little difference in practice and was difficult to integrate with second-order preconditioners like SPRING (Goldshlager et al., 2024).

## G. Normalizing constant ratio estimation

The MSIS estimator requires the ratios $\kappa_s = \frac{\mathcal{Z}_1^2}{\mathcal{Z}_s^2}$ for $s \in \{2, \ldots, N_s\}$. We estimate these via the iterative bridge sampling algorithm of Meng & Wong (1996).

**Bridge sampling.** Given unnormalized densities $q_s = \psi_s^2$ and $q_t = \psi_t^2$ with unknown normalizing constants, bridge sampling estimates their ratio by introducing a *bridge function* $\alpha(\mathbf{r})$ that connects expectations under both distributions. The key identity is

$$\kappa_s \mathbb{E}_{\rho_s}[\alpha \, q_t] = \kappa_t \mathbb{E}_{\rho_t}[\alpha \, q_s], \tag{47}$$

which holds for any $\alpha > 0$ and relates the ratio $\frac{\kappa_s}{\kappa_t}$ to computable expectations. The optimal bridge function that minimizes estimator variance is $\alpha^* \propto \frac{1}{\kappa_s q_s + \kappa_t q_t}$, which places more weight where both densities have support.

**Multi-state estimation.** For $N_s$ states, we use a shared bridge function $\alpha = \frac{1}{q_{\text{mix}}^{(\boldsymbol{\kappa})}}$ with $q_{\text{mix}}^{(\boldsymbol{\kappa})} = \frac{1}{N_s}\sum_s \kappa_s q_s$. Defining responsibilities $w_s = \frac{q_s}{N_s q_{\text{mix}}^{(\boldsymbol{\kappa})}}$, the identity becomes $\kappa_s \mathbb{E}_{\rho_s}[w_t] = \kappa_t \mathbb{E}_{\rho_t}[w_s]$. Using all $N_s(N_s - 1)$ pairwise identities with $\kappa_1 = 1$ yields a linear system $\boldsymbol{B}\boldsymbol{\kappa} = \boldsymbol{b}$ for $\boldsymbol{\kappa} = (\kappa_2, \ldots, \kappa_{N_s})^T$:

$$B_{st} = \begin{cases} \sum_{s' \neq s} \mathbb{E}_{\rho_{s'}}[w_s] & \text{, if } s = t, \\ -\mathbb{E}_{\rho_s}[w_t] & \text{, else,} \end{cases} \tag{48}$$

$$b_s = \mathbb{E}_{\rho_s}[w_1]. \tag{49}$$

Since $w_s$ depends on the unknown $\boldsymbol{\kappa}$ through $q_{\text{mix}}^{(\boldsymbol{\kappa})}$, we iterate: initialize $\hat{\boldsymbol{\kappa}}^{(0)} = \mathbf{1}$, then solve $\hat{\boldsymbol{\kappa}}^{(t+1)} = (\boldsymbol{B}^{(t)})^{-1}\boldsymbol{b}^{(t)}$. We run 10 iterations per optimization step and clip updates to a factor of 2 for stability.

### G.1. Computational cost of bridge sampling

We have found the impact of the normalizing constant ratio estimation on step time to be negligible. To confirm, we have measured the execution time of the iterative algorithm on an Nvidia H100 accelerator for a range of configurations. The step-time contribution is shown in Tab. 1. At 16 states and 8 molecules, the algorithm takes around $0.83\,\text{ms}$, which is less than $0.015\,\%$ of the VMC step time for a comparable configuration.

*Table 1.* Step-time contribution of the iterative bridge sampling algorithm measured on an Nvidia H100 for state counts $N_{\rm s}$ and molecule counts $N_{\rm mol}$. The total batch size is kept constant at 4096. Times are given in ms.

| $N_{\rm mol}$ / $N_{\rm s}$ | 1 | 2 | 4 | 8 | 16 | 32 | 64 |
|---|---|---|---|---|---|---|---|
| 1 | 0.040 | 0.082 | 0.080 | 0.081 | 0.085 | 0.084 | 0.085 |
| 2 | 0.790 | 0.675 | 0.617 | 0.583 | 0.539 | 0.555 | 0.518 |
| 4 | 0.819 | 0.649 | 0.614 | 0.587 | 0.579 | 0.587 | 0.567 |
| 8 | 0.848 | 0.693 | 0.684 | 0.670 | 0.664 | 0.671 | 0.731 |
| 16 | 0.996 | 0.815 | 0.812 | 0.829 | 0.900 | 0.885 | 0.894 |
| 32 | 1.262 | 1.251 | 1.306 | 1.352 | 1.379 | 1.410 | 1.421 |
| 64 | 1.872 | 2.526 | 2.573 | 2.656 | 2.764 | 2.868 | 2.782 |

## H. Memory usage of separate orbitals per state

A naive extension to multiple states expands the orbital matrices with a state dimension, i.e., $\Phi_{skip}$, requiring separate orbital weights $w_{skp}$ and envelopes $\chi_{skp}$ for each state (Schätzle et al., 2025; Pfau et al., 2024). While conceptually simple, this approach leads to prohibitive memory requirements when combined with approximate second-order optimizers such as SPRING (Goldshlager et al., 2024), which require storing the Jacobian of the network outputs with respect to parameters.

For typical hyperparameters, batch size $N_{\rm b} = 4096$, $N_{\rm orb/nuc} = 8$ orbitals per nucleus, embedding dimension $N_{\rm f} = 256$, $N_{\rm k} = 16$ determinants, $N_{\rm s} = 32$ states, and meta-network embedding dimension of 64, the Jacobian of the linear layer mapping nuclear embeddings to orbital projections alone requires

$$4096 \times 8 \times 256 \times 16 \times 32 \times 64 \times 4 \,\text{bytes} \approx 1 \,\text{TB} \tag{50}$$

of memory, far exceeding current GPU capacities.

Previous works have addressed this limitation differently. Schätzle et al. (2025) compute and precondition gradients for each state separately, then average the updates. However, generalized wave function methods have demonstrated that jointly preconditioning the total loss yields better optimization dynamics (Gao & Günnemann, 2022; Scherbela et al., 2024). In Pfau et al. (2024), this issue does not arise because they use fewer orbitals, treat orbitals as direct parameters, and do not employ a meta-network for generalization.

Our Excited Pfaffian architecture avoids this memory bottleneck entirely by sharing orbitals across states and varying only the lightweight antisymmetrizer $A_{sk}$, whose parameter count is independent of the embedding dimension.

## I. Comparison to naive multi-state architecture

App. H establishes that the naive multi-state extension is incompatible with second-order preconditioners at our standard scale ($N_{\rm k} = 16$). For a direct empirical comparison nonetheless, we reduce the determinant count to $N_{\rm k} = 1$, the largest configuration that fits in H100 memory for the naive architecture on 33-state beryllium. Even at this reduced scale, the naive architecture requires $48\,\text{s}$ per step, compared to $7.2\,\text{s}$ for an Excited Pfaffian at $N_{\rm k} = 1$ and $8\,\text{s}$ for our standard $N_{\rm k} = 16$ Excited Pfaffian. Fig. 12 plots the convergence of the total energy. Per step, the $N_{\rm k} = 1$ Excited Pfaffian and the naive $N_{\rm k} = 1$ architecture follow nearly identical trajectories, confirming that our parametrization preserves expressivity at equal capacity. Per wall-clock hour, the standard $N_{\rm k} = 16$ Excited Pfaffian reaches within $\sim 10^{-5}\,E_{\rm h}$ of its converged energy in under $40\,\text{h}$, while the naive architecture is still $\sim 2$ orders of magnitude above this after $80\,\text{h}$. We estimate that full convergence of the naive architecture would require $\sim 111$ A100 GPU-days, exceeding our compute budget.

## J. Pretraining target construction

Pretraining neural-network wave functions to Hartree-Fock solutions reduces the number of costly VMC optimization steps (Hermann et al., 2023), as HF captures approximately 99% of the total energy (Szabo & Ostlund, 2012). However, constructing consistent pretraining targets for multi-structure excited states poses unique challenges: HF molecular orbital coefficients are only defined up to unitary transformations, and different structures yield independently computed solutions with arbitrary relative phases and orderings. We address this by (1) organizing structures into a directed acyclic graph, (2)

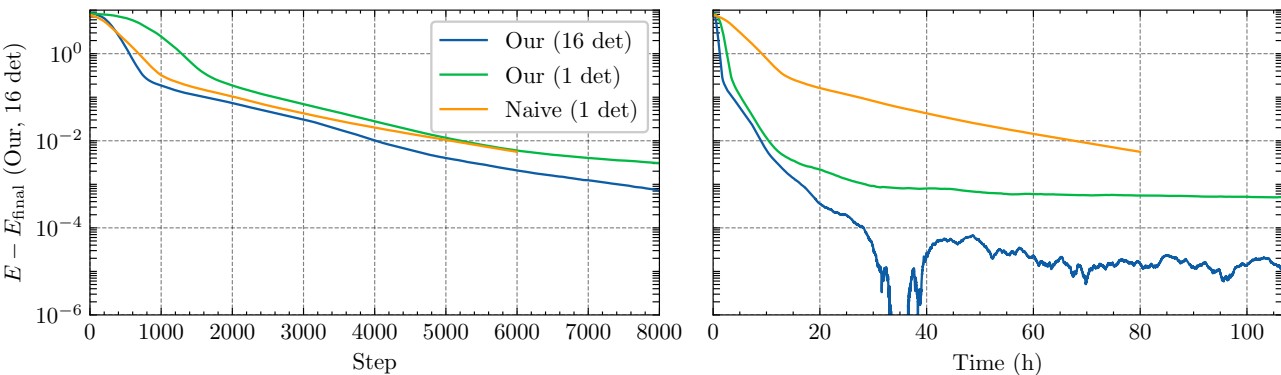

*Figure 12.* Convergence of the total energy on 33-state Be for the standard Excited Pfaffian ($N_{\mathrm{k}} = 16$), a single-determinant Excited Pfaffian ($N_{\mathrm{k}} = 1$), and the naive multi-state architecture at $N_{\mathrm{k}} = 1$ (the largest that fits in memory). Left: per training step. Right: per wall-clock hour. Energies are reported relative to the converged energy of the standard $N_{\mathrm{k}} = 16$ Excited Pfaffian.

selecting consistent excitations at the root, and (3) propagating canonical orbitals between structures via basis projection and overlap maximization.

**Structure graph construction.** To ensure smooth pretraining targets across structures, we must establish a canonical correspondence between molecular orbitals at different geometries. Naively running independent HF calculations at each structure yields orbitals with arbitrary phases and orderings, causing discontinuous pretraining targets that hinder generalization.

We observe that geometrically similar structures have similar electronic structure, so their HF orbitals can be aligned with small error. This motivates organizing structures into a graph where edges connect similar geometries. Specifically, we compute pairwise RMSD between all structures sharing the same nuclei $\mathbf{Z}$ and electronic configuration $(N_\uparrow, N_\downarrow)$, and construct a minimum spanning tree (MST) over this complete graph. The MST minimizes the total edge weight, which corresponds to minimizing the cumulative geometric dissimilarity along propagation paths.

The MST defines connectivity but not direction. To obtain a rooted tree suitable for sequential propagation, we select the root as the node minimizing the tree's diameter (maximum distance to any leaf). This choice minimizes the worst-case propagation depth, reducing error accumulation for the most distant structures. The rooted tree naturally defines a DAG via parent-child relationships.

Traversing this DAG in topological order yields a sequence where each non-root structure $i$ has a unique predecessor $p(i)$ whose orbitals have already been determined. This reduces the global $N$-structure matching problem to $N-1$ independent pairwise matchings, each between geometrically adjacent structures where alignment is most reliable.

**Excitation selection.** Following Pfau et al. (2024), we enumerate excited states via orbital selectors $\Pi_s \in \{0, 1\}^{N_{\mathrm{o}} \times N_{\mathrm{e}}}$ that specify which orbitals to occupy for each state $s$. To maintain consistency across structures, we determine the excitation pattern (i.e., which occupied orbitals are replaced by which virtual orbitals) at the root structure and apply the same $\Pi_s$ to all related structures. This ensures that corresponding excited states across different geometries represent the same electronic transition, enabling smooth interpolation of potential energy surfaces. However, this requires that molecular orbitals maintain consistent ordering and character across structures, which we address in the following paragraphs.

**Orbital propagation.** We perform a single Hartree-Fock computation at the root structure, obtaining basis functions $\Xi_{\mathrm{root}}$ and molecular orbital coefficients $C_{\mathrm{root}}$. For each non-root structure $i$ with predecessor $p(i)$, we propagate the orbitals by projecting $C_{p(i)}$ onto structure $i$'s basis $\Xi_i$. The projection must preserve orthonormality of the molecular orbitals. Given the overlap matrix $\tilde{O}_i$ of structure $i$'s basis functions, where $(\tilde{O}_i)_{\mu\nu} = \langle \xi_\mu | \xi_\nu \rangle$, we compute:

$$C_i = C_{p(i)} \left( C_{p(i)}^T \tilde{O}_i C_{p(i)} \right)^{-\frac{1}{2}}. \tag{51}$$

This symmetric orthogonalization ensures $C_i^T \tilde{O}_i C_i = I$, preserving orthonormality while staying maximally close to the predecessor's orbitals in a least-squares sense.

**Block-wise overlap maximization.** To ensure consistent orbital ordering, we seek an orthogonal transformation $Q_i \in O(N_o)$ that minimizes the distance between the molecular orbitals of structure $i$ and its predecessor $p(i)$:

$$Q_i^* = \arg \min_{Q_i \in O(N_o)} \left\| \Xi_i C_i Q_i - \Xi_{p(i)} C_{p(i)} \right\|_F^2. \tag{52}$$

Expanding the Frobenius norm and using the orthonormality of molecular orbitals, this simplifies to maximizing $\text{tr}\left(Q_i^T \bar{O}_i\right)$ where $\bar{O}_i = C_i^T \langle \Xi_i | \Xi_{p(i)} \rangle C_{p(i)}$ is the inter-structure molecular orbital overlap matrix. This is an orthogonal Procrustes problem with closed-form solution $Q_i^* = \hat{U}_i \hat{V}_i^T$, where $\hat{U}_i \hat{\Sigma}_i \hat{V}_i^T = \bar{O}_i$ is the singular value decomposition. We then update $C_i \leftarrow C_i Q_i^*$ to obtain the aligned orbitals.

However, applying this global alignment over all orbitals can produce spurious matchings. Core orbitals have small spatial extent and may have higher overlap with spatially extended higher-energy orbitals from nearby atoms than with their true counterparts. To address this, we partition orbitals into groups based on their Hartree-Fock orbital energies $\epsilon_k$ (Szabo & Ostlund, 2012) and solve independent Procrustes problems within each group. This block-wise approach is physically motivated: orbitals with degenerate or near-degenerate energies are not uniquely defined and may mix arbitrarily, so matching within energy shells respects this gauge freedom.

We cluster orbitals using kernel density estimation on the orbital energies. Given energies $\{\epsilon_k\}_{k=1}^{N_o}$, we estimate the density using a Gaussian kernel $\hat{f}(\epsilon) = \frac{1}{N_o h \sqrt{2\pi}} \sum_k \exp\left(-\frac{(\epsilon - \epsilon_k)^2}{2h^2}\right)$ with bandwidth $h$. Local maxima of $\hat{f}$ define cluster centroids, and each orbital is assigned to its nearest centroid. This approach has a single hyperparameter (the bandwidth $h$) and naturally adapts to the energy level structure of each system.

## K. Carbon dimer comparison

Fig. 13 provides a side-by-side comparison of $C_2$ potential energy surfaces computed by Excited Pfaffian, NES (Pfau et al., 2024), and Schätzle et al. (2025), all benchmarked against SHCI reference calculations. To improve readability, we plot singlet states (left column) and triplet states (right column) separately, with each row corresponding to a single method. As we are interested in relative energies, all SHCI surfaces are normalized to the lowest energy of the respective method.

Excited Pfaffian closely tracks the SHCI baseline across the full bond-length range from $0.995\,\text{Å}$ to $1.87\,\text{Å}$. The curves maintain correct state ordering and accurate energy gaps throughout, including the challenging stretched-geometry regime where electron correlation is strongest, with two exceptions. We observe the singlet state swap between the $X^1\Sigma_g^+$ and $B'^1\Sigma_g^+$ discussed in the main body, and a state swap between one of the degeneracies in $A^1\Pi_u$ and $B^1\Delta_g$.

Analyzing the singlet state tracking in stretched geometries, Schätzle et al. (2025) suggest a state swap between the $X^1\Sigma_g^+$ and $A^1\Pi_u$ for their PES. We offer a different explanation for the lost state tracking. Noting that the $X^1\Sigma_g^+$ of Schätzle et al. (2025) follows a similar trajectory as in the Excited Pfaffian PES, we suggest that the state starts tracking $B'^1\Sigma_g^+$ at longer bond lengths. The $B^1\Delta_g$ state is twofold degenerate, but only captured once by Schätzle et al. (2025). Hence, one of the two degenerate $A^1\Pi_u$ states drops to the lower $B^1\Delta_g$ at bond lengths larger than $\sim 1.6\,\text{Å}$. Both states track the lost $X^1\Sigma_g^+$ surface, leading to the omission of the lowest singlet PES at stretched geometries. This illustrates the need to compute larger numbers of excited states, as made possible by Excited Pfaffian. In the triplet spin sector, the method of Schätzle et al. (2025) exhibits visible deviations, even in low-lying states, at longer bond lengths ($> 1.4\,\text{Å}$).

The single-point energies computed by NES closely match the SHCI baseline in all geometries across both spin sectors. Since the NES surfaces were computed in single-point calculations, automatic state tracking is not possible there.

Notably, Excited Pfaffian achieves this accuracy using 800M samples compared to 32B for NES and 3.2B for Schätzle et al. (2025), representing a $\sim 40\times$ and $\sim 4\times$ reduction in sample cost, respectively.

## L. Ethylene state swap

As discussed by Barbatti et al. (Barbatti et al., 2004), the torsion region between $15°$ and $30°$ involves a crossing between a Rydberg state and the valence excited state, resulting in a change of electronic character of the upper state. We hypothesize that the Rydberg state's diffuse excited electron contributes little to the valence electron density, potentially making it appear more similar to the ground state in the bonding region than to the valence excited state. As our model favors parameter closeness, this similarity may cause the observed swap, illustrated in Fig. 14. Increasing the number of modeled states to

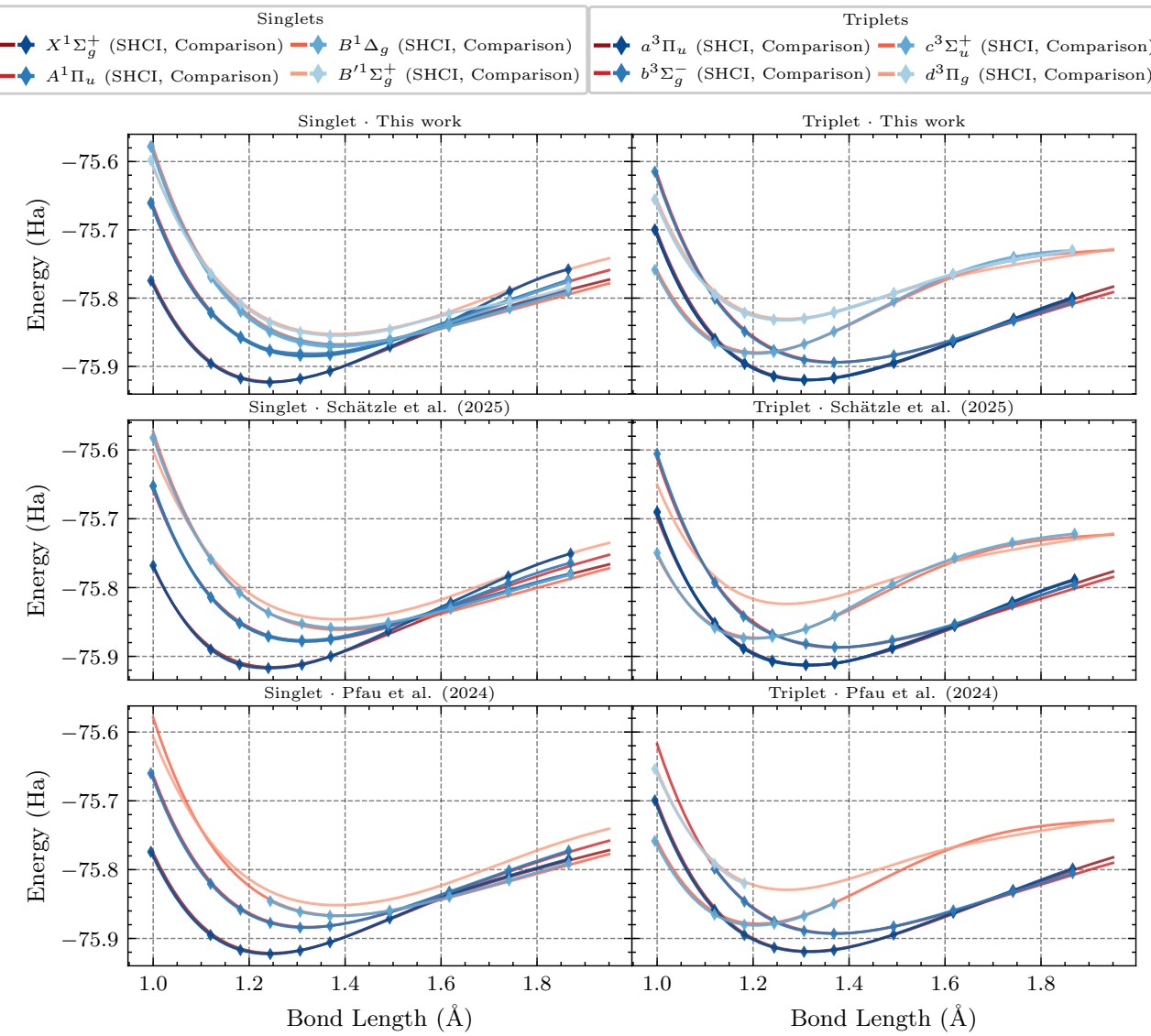

*Figure 13.* Comparison of $C_2$ potential energy surfaces across methods. Each row shows singlet (left) and triplet (right) states for Excited Pfaffian (top), the method of Schätzle et al. (2025) (middle), and NES (Pfau et al., 2024) (bottom). For Excited Pfaffian and the method of Schätzle et al. (2025), we give the raw energies before correcting state swaps and averaging degeneracies. Raw energies for NES were not available. SHCI reference curves are shown in red/orange; method results are in blue. All three methods capture the qualitative PES structure, but the method of Schätzle et al. (2025) shows increased deviation from the SHCI baseline (Holmes et al., 2017) at stretched geometries.

explicitly include the Rydberg state, as well as improved state-tracking methods, may resolve this issue. We leave a detailed investigation to future work.

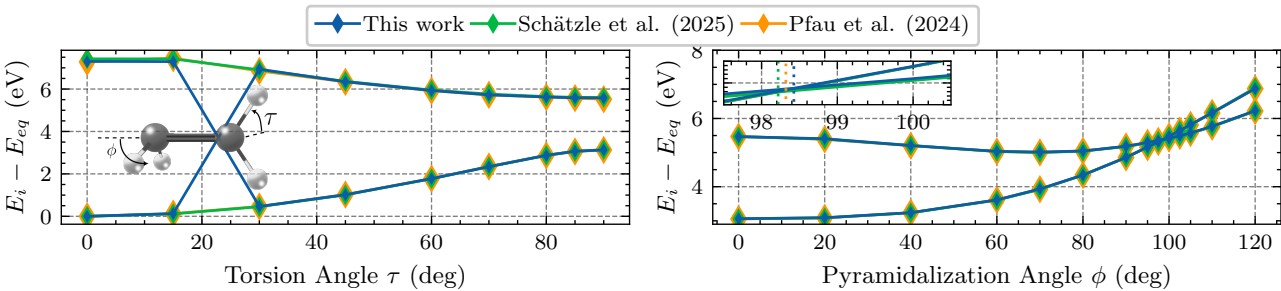

*Figure 14.* Potential energy surfaces of ethylene along torsion ($\tau$) and pyramidalization ($\phi$). The Excited Pfaffian PES for both slices is computed in a single neural wave function. Left: rotation about the C–C bond. Right: pyramidalization showing a state crossing $\phi \approx 99°$ (inset).

## M. Additional Bhattacharyya coefficients

In addition to the unsigned wave function overlaps given in Fig. 8, we provide the Bhattacharyya coefficients of the second-row atoms in Fig. 15, and all 33 states of beryllium in Fig. 16.

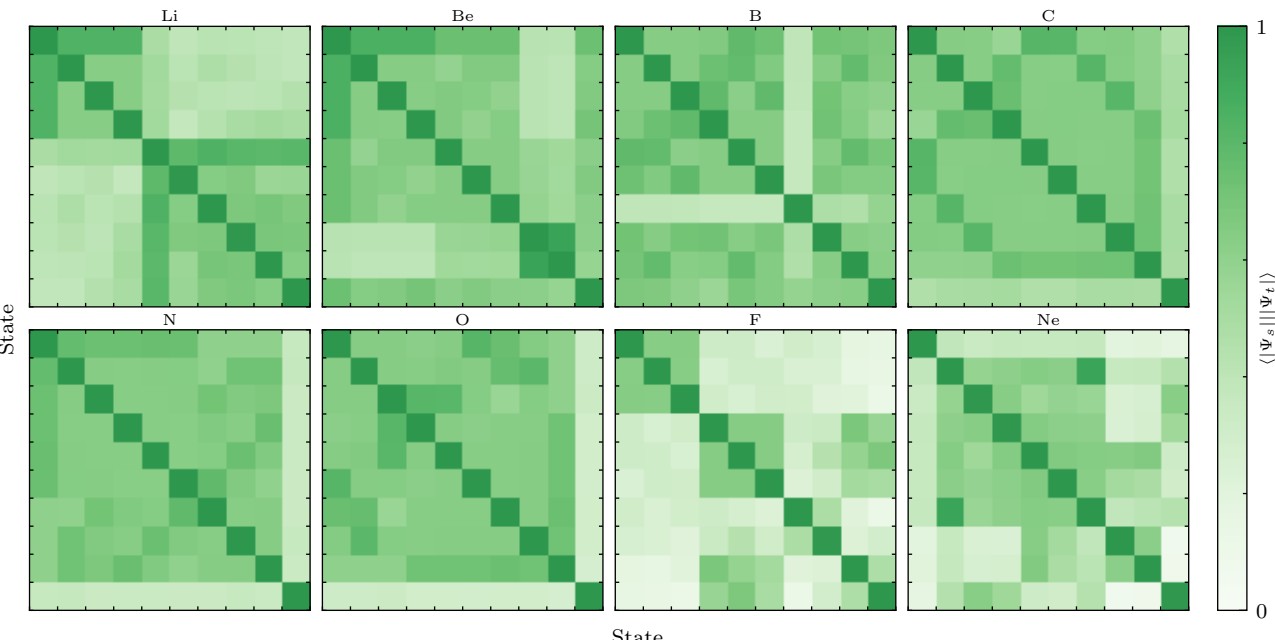

*Figure 15.* Spatial overlap of atomic excited states measured as Bhattacharyya coefficients $\langle |\Psi_s| \| |\Psi_t| \rangle$.

## N. Empirical step-time comparison

We compare per-step training cost against NES (Pfau et al., 2024) and Szabó et al. (2024) through direct wall-clock measurements on Nvidia A100 GPUs. Fig. 17 visualizes the resulting scaling with $N_s$ on Be and $C_2$, where Excited Pfaffian stays near-flat while NES grows roughly quadratically. Tab. 2 consolidates total training sample counts and shows that Excited Pfaffian uses 2–80× fewer samples than each baseline. Tables 3 and 4 report per-step training times for the second-row atoms and the 12 molecules of Fig. 4, while Tab. 5 covers the $C_2$ PES (Fig. 5) and the ethylene PES (Fig. 7). For Adam and KFAC, step times are measured at 1000 walkers and scaled to 4000 walkers. SPRING step times are measured directly at 4000 walkers. Since Excited Pfaffian trains a single model across all structures, we compare against the cumulative step time of per-structure baseline training, yielding 7–208× speedups depending on system and optimizer. The SPRING preconditioner contributes an outsized overhead on small systems (16.8× over Adam on atoms) that shrinks

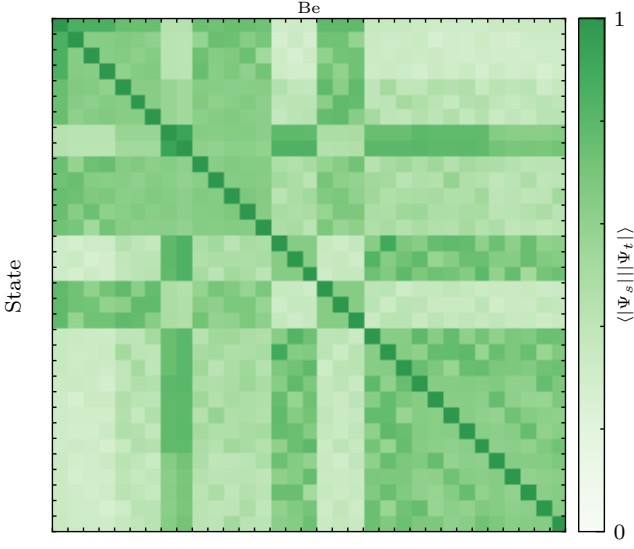

*Figure 16.* Spatial overlap of 33 states of beryllium measured as Bhattacharyya coefficients $\langle|\Psi_s|\|\Psi_t|\rangle$.

with system size ($2.7\times$ on ethylene), as the cost of the rest of the VMC optimization grows faster than the cost of the preconditioner.

*Table 2.* Overview of total sample counts used in training in different experiments across works and improvement range for this work.

| Method | Atoms | Molecules | $C_2$ PES | Ethylene PES |
|---|---|---|---|---|
| Pfau et al. (2024) | 64B | 48B | 32B | 27.6B |
| Szabo et al. (2024) | 16B | 12B | 8B | 9.2B |
| Schätzle et al. (2025) | - | - | 3.2B | 1.6B |
| Ours | 800M | 800M | 800M | 800M |
| Sample reduction range | 20–80× fewer | 15–60× fewer | 4–40× fewer | 2–30× fewer |

### N.1. NES at equal compute

The per-step measurements above quantify the speed advantage. We now verify that this advantage translates into a tractability gap. We allocate the same total compute we used for the 12-state $C_2$ PES (10 geometries, 200k steps, $\sim 11$ s/step) to NES (Pfau et al., 2024), which trains one model per geometry. At NES's per-step cost of $\sim 50$ s, this budget supports only $\sim$4.5k optimization steps per geometry, far below the 200k steps NES typically requires. NES additionally required 100k pretraining steps to avoid NaNs at initialization, while 1k pretraining steps sufficed for our method, so this comparison favors NES by excluding pretraining from the budget. Pfau et al. (2024) only train 8 states (ground state plus S1–S7) on $C_2$. We trained NES for 12 states to match our setup but found the optimization to be very unstable. Tab. 6 reports the resulting excitation energies on $C_2$ at 0.995 Å. At full convergence, NES reproduces our ground-state excitation to within $10^{-4}$ eV, cross-validating both methods on this benchmark. At the equal-compute budget, however, NES misses the ground-state excitation by $\sim 0.95$ eV and overestimates higher states by up to $\sim 9.6$ eV. The per-step speedup therefore translates directly into accuracy at fixed compute, and NES cannot reach competitive energies within our budget.

## O. Experimental setup

### O.1. Hyperparameters

The default hyperparameters used in all experiments are listed in Tab. 7. Most hyperparameters are directly taken from previous works (Gao & Günnemann, 2024; Szabó et al., 2024). For the atomic systems in Fig. 4, we used 16 orbitals per nucleus and disabled the snap spin penalty. Otherwise, the snap spin penalty was used in every experiment, except for the

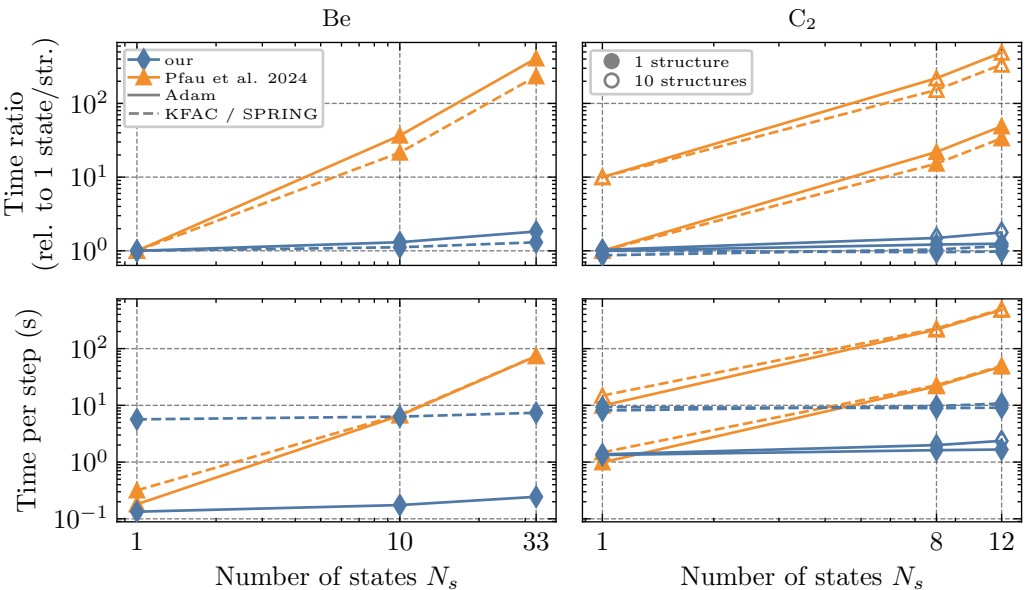

*Figure 17.* Per-step training time as a function of the number of states $N_s$ on Be (left; 1, 10, 33 states) and $C_2$ (right; 1, 8, 12 states). Top: time normalized to the single-state-per-structure cost. Bottom: absolute time per step. Excited Pfaffian (blue) and NES (Pfau et al., 2024) (orange) are compared with Adam (solid) and their native second-order preconditioner (dashed; SPRING for ours, KFAC for NES). For $C_2$, filled markers show the per-structure cost. Hollow markers show the cumulative cost across the 10 dissociation geometries (one shared model for Excited Pfaffian, one model per geometry for NES). Excited Pfaffian maintains near-flat scaling, while NES grows roughly quadratically with $N_s$.

*Table 3.* Measured training step time in s for 10 states of atomic systems. Measurements for Adam and KFAC are taken for 1000 walkers and up-scaled to 4000 walkers. Measurements for SPRING are directly taken for 4000 walkers.

| Method | Optimizer | Li | Be | B | C | N | O | F | Ne | $\sum$ | $\frac{\sum}{\sum_{\text{our}}}$ |
|---|---|---|---|---|---|---|---|---|---|---|---|
| Pfau et al. (2024) | Adam | 5.4 | 6.7 | 9.7 | 11.8 | 14.7 | 16.6 | 22.3 | 25.6 | 112.7 | 113× |
|  | KFAC | 5.8 | 7.1 | 10.0 | 12.2 | 15.2 | 17.1 | 22.8 | 26.2 | 116.4 | 7× |
| Szabo et al. (2024) | Adam | 3.3 | 3.8 | 4.3 | 5.8 | 7.0 | 7.6 | 9.3 | 10.1 | 51.2 | 51× |
|  | KFAC | 11.4 | 12.4 | 13.2 | 13.7 | 15.4 | 16.4 | 18.2 | 18.8 | 119.5 | 7× |
| This Work | Adam | - | - | - | - | - | - | - | - | **1.0** | 1× |
|  | SPRING | - | - | - | - | - | - | - | - | **16.8** | 1× |

*Table 4.* Measured training step time in s for 5 states of molecular systems. Measurements for Adam and KFAC are taken for 1000 walkers and up-scaled to 4000 walkers. Measurements for SPRING are directly taken for 4000 walkers.

| Method | Optimizer | BH | HCl | $H_2O$ | $H_2S$ | BF | CO | $C_2H_4$ | $CH_2O$ | $CH_2S$ | HNO | HCF | $H_2CSi$ | $\sum$ | $\frac{\sum}{\sum_{\text{our}}}$ |
|---|---|---|---|---|---|---|---|---|---|---|---|---|---|---|---|
| Pfau et al. (2024) | Adam | 3.5 | 4.9 | 7.3 | 4.8 | 12.3 | 12.3 | 14.8 | 14.7 | 12.4 | 14.6 | 14.6 | 9.4 | 125.4 | 32× |
|  | KFAC | 3.9 | 5.2 | 7.7 | 5.2 | 12.8 | 12.8 | 15.4 | 15.4 | 13.0 | 15.2 | 15.2 | 9.9 | 131.7 | 9× |
| Szabo et al. (2024) | Adam | 3.4 | 15.2 | 5.6 | 15.2 | 9.2 | 9.4 | 10.6 | 10.4 | 24.8 | 10.4 | 11.2 | 21.2 | 146.6 | 37× |
|  | KFAC | 7.4 | 19.6 | 9.4 | 19.6 | 13.2 | 13.2 | 15.2 | 15.0 | 29.8 | 15.0 | 15.0 | 25.6 | 198.0 | 14× |
| This Work | Adam | - | - | - | - | - | - | - | - | - | - | - | - | **3.9** | 1× |
|  | SPRING | - | - | - | - | - | - | - | - | - | - | - | - | **14.0** | 1× |

*Table 5.* Measured training step time in s for the PES reported in the main body. Measurements for Adam and KFAC are taken for 1000 walkers and up-scaled to 4000 walkers. Measurements for SPRING are directly taken for 4000 walkers.

| Method | Optimizer | C$_2$ | | | Ethylene | | |
| --- | --- | --- | --- | --- | --- | --- | --- |
| | | Step time | Total 10 structures | $\frac{total}{our\ total}$ | Step time | Total 23 structures | $\frac{total}{our\ total}$ |
| Pfau et al. (2024) | Adam | 49.0 | 490.4 | 208× | 3.9 | 89.2 | 24× |
| | KFAC | 49.9 | 499.2 | 46× | 4.2 | 97.5 | 9× |
| Szabo et al. (2024) | Adam | 14.0 | 140.0 | 59× | 5.5 | 125.6 | 33× |
| | KFAC | 24.8 | 248.0 | 23× | 7.4 | 170.2 | 16× |
| This Work | Adam | - | **2.4** | 1× | - | **3.8** | 1× |
| | SPRING | - | **11.0** | 1× | - | **10.4** | 1× |

*Table 6.* Excitation energies (eV) for C$_2$ at a bond distance of 0.995 Å. "NES (paper)" lists the values reported in Pfau et al. (2024), who only report energies averaged over degenerate states (entries repeated within each degeneracy here). "NES (4.5k)" is NES retrained from scratch at the per-geometry compute budget equivalent to our 12-state, 10-geometry C$_2$ PES.

| Method | S1 | S2 | S3 | S4 | S5 | S6 | S7 | S8 | S9 | S10 | S11 |
| --- | --- | --- | --- | --- | --- | --- | --- | --- | --- | --- | --- |
| Ours | 0.4294 | 2.0161 | 2.0438 | 3.0738 | 3.1192 | 3.2308 | 3.2352 | 4.3443 | 4.7957 | 5.3375 | 5.3634 |
| NES (paper) | 0.4293 | 2.0369 | 2.0369 | 3.1051 | 3.1051 | 3.2752 | 3.2752 | — | — | — | — |
| NES (4.5k) | 1.3817 | 3.5633 | 3.7173 | 4.4938 | 4.7929 | 5.1994 | 6.6696 | 12.7497 | 14.1597 | 14.6219 | 15.0104 |

carbon dimer PES and the ethylene PES, where the spin states were enforced explicitly with the $S^+$ penalty. Pretraining was only necessary for the carbon dimer PES; all other experiments were run without pretraining.

### O.2. Source code

Our implementation is based on the source code of Neural Pfaffians (Gao & Günnemann, 2024). It is publicly available on GitHub[1].

### O.3. Computational details

We report the total compute times for all experiments in Tab. 8. Most experiments were run on 1-4 Nvidia A100 or Nvidia H100 accelerators, by parallelizing across the sample dimension. We convert H100 GPU-hours to A100 GPU-hours with a factor of 2.

The VMC code is implemented in JAX (Bradbury et al., 2018) and neural networks are implemented using Flax (Heek et al., 2024). HF pretraining targets are computed using PySCF (Sun et al., 2018). For gradient updates, we employ the optimizer utilities provided by optax (DeepMind et al., 2020). For second-order optimization, i.e., gradient preconditioning, we use SPRING (Goldshlager et al., 2024). Preconditioning auxiliary gradients, such as the $S^+$ gradients, is not trivially supported by SPRING. Following Scherbela et al. (2025), we split the gradient into a component inside the span of the Jacobian used in preconditioning, and one orthogonal component. Only the in-span component is preconditioned.

## P. Numerical results

In tables 9 to 13 we provide numerical results for the experimental results presented in Sec. 5.

---

[1] https://github.com/n-gao/neural-pfaffian

*Table 7.* Hyperparameters used in all experiments.

| Category | Parameter | Value |
|---|---|---|
| *Wave Function Architecture* | | |
| | Embedding network | Moon |
| | Hidden dimension | 256 |
| | Number of layers | 4 |
| | Envelopes per nucleus | 8 |
| | Orbitals per nucleus and determinant | 8 |
| | Number of determinants | 16 |
| | Jastrow factor | MLP $[128, 32]$ + cusp |
| *Meta Network* | | |
| | Message dimension | 32 |
| | Embedding dimension | 64 |
| | Number of layers | 3 |
| | Radial basis functions | 6 |
| *VMC Optimization* | | |
| | Training epochs | 200,000 |
| | MCMC steps per epoch | 20 |
| | Target acceptance rate | 0.525 |
| | Preconditioner | SPRING |
| | Damping | $10^{-3}$ |
| | Learning rate schedule | $\frac{0.02}{1+\frac{t}{10000}}$ |
| | Gradient norm clipping | 0.032 |
| | Local energy clipping | $5\times$ (95th percentile) |
| *Overlap Penalty* | | |
| | Penalty scale | 4.0 |
| | Clipping | $5\times$ (95th percentile) |
| *Spin Penalty* | | |
| | $S^+$ Penalty scale | 8.0 |
| | Snap Penalty Scale | $0.1\,\mathrm{sigmoid}\left(\frac{t-100,000}{10,000/2\log(9.0)}\right)$ |
| | Clipping | $5\times$ (95th percentile) |
| | Masking | $10\times$ (95th percentile) |
| *Pretraining* | | |
| | Epochs | 2,000 |
| | Basis set | aug-cc-pVTZ |
| | Optimizer | LAMB |
| | Sampling distribution | Neural network |
| | MCMC steps per epoch | 20 |
| | Learning rate schedule | $\frac{0.001}{1+\frac{t}{1000}}$ |
| *Evaluation* | | |
| | Samples per energy | $10^6$ |
| | MCMC steps | 100 |

*Table 8.* Total compute times. Nvidia H100 GPU-hours correspond to 2 Nvidia A100 GPU-hours.

| Experiment | A100 GPU-hours |
|---|---|
| Atomic excitations (Fig. 4, left) | 640 |
| Molecules (Fig. 4, right) | 792 |
| Carbon dimer PES (Fig. 5) | 600 |
| Ethylene PES (Fig. 7) | 640 |
| 33 states of Be (Fig. 6) | 231 |

*Table 9.* Energies and excitation energies of the molecules in Fig. 4 (right) obtained from the jointly trained Excited Pfaffian in atomic units.

| Molecule | Energy (Ha) | $\Delta E$ (Ha) | Molecule | Energy (Ha) | $\Delta E$ (Ha) |
|---|---|---|---|---|---|
| BH | -25.28897(8) | - | $C_2H_4$ | -78.5842(2) | - |
| | -25.24029(8) | 0.04868(12) | | -78.4171(2) | 0.1671(3) |
| | -25.24021(8) | 0.04876(12) | | -78.3136(2) | 0.2706(3) |
| | -25.18341(9) | 0.10555(12) | | -78.3134(2) | 0.2708(3) |
| | -25.18336(9) | 0.10561(12) | | -78.2867(2) | 0.2975(3) |
| HCl | -15.5959(4) | - | $CH_2O$ | -114.5035(3) | - |
| | -15.32415(18) | 0.2717(4) | | -114.3709(3) | 0.1326(4) |
| | -15.3239(2) | 0.2719(4) | | -114.3565(3) | 0.1470(4) |
| | -15.3056(2) | 0.2903(4) | | -114.2801(3) | 0.2234(4) |
| | -15.3054(2) | 0.2905(5) | | -114.2360(4) | 0.2675(5) |
| $H_2O$ | -76.4366(2) | - | $CH_2S$ | -49.45841(19) | - |
| | -76.1663(2) | 0.2703(3) | | -49.38641(19) | 0.0720(3) |
| | -76.1529(3) | 0.2837(3) | | -49.37654(19) | 0.0819(3) |
| | -76.0909(2) | 0.3458(3) | | -49.3326(2) | 0.1258(3) |
| | -76.0903(2) | 0.3463(3) | | -49.2406(2) | 0.2178(3) |
| $H_2S$ | -11.38833(11) | - | HNO | -130.4770(4) | - |
| | -11.17652(11) | 0.21181(15) | | -130.4433(4) | 0.0337(5) |
| | -11.17016(14) | 0.21817(18) | | -130.4132(4) | 0.0638(5) |
| | -11.16364(12) | 0.22469(16) | | -130.3178(3) | 0.1591(5) |
| | -11.15668(15) | 0.23165(18) | | -130.2705(4) | 0.2065(5) |
| BF | -124.6739(3) | - | HCF | -138.4130(3) | - |
| | -124.5397(3) | 0.1343(5) | | -138.3780(4) | 0.0350(5) |
| | -124.5396(3) | 0.1343(5) | | -138.3209(4) | 0.0921(5) |
| | -124.4385(4) | 0.2354(5) | | -138.2015(4) | 0.2114(5) |
| | -124.4384(3) | 0.2355(5) | | -138.1613(4) | 0.2517(5) |
| CO | -113.3208(3) | - | $H_2CSi$ | -43.10937(20) | - |
| | -113.0885(3) | 0.2323(5) | | -43.03432(19) | 0.0751(3) |
| | -113.0881(3) | 0.2327(4) | | -43.03430(18) | 0.0751(3) |
| | -113.0060(3) | 0.3148(4) | | -43.01645(19) | 0.0929(3) |
| | -113.0044(3) | 0.3164(4) | | -43.00067(19) | 0.1087(3) |

*Table 10.* Energies and excitation energies of the atoms in Fig. 4 (left) obtained from the jointly trained Excited Pfaffian in atomic units.

| Atom | Energy (Ha) | $\Delta E$ (Ha) | Atom | Energy (Ha) | $\Delta E$ (Ha) |
|---|---|---|---|---|---|
| Li | -7.47807(3) | - | N | -54.58886(14) | - |
|  | -7.41013(4) | 0.06794(5) |  | -54.50060(14) | 0.0883(2) |
|  | -7.41005(4) | 0.06802(5) |  | -54.50056(15) | 0.0883(2) |
|  | -7.40994(4) | 0.06813(5) |  | -54.50055(14) | 0.0883(2) |
|  | -7.35424(3) | 0.12382(4) |  | -54.50033(15) | 0.0885(2) |
|  | -7.33542(9) | 0.14264(9) |  | -54.50033(15) | 0.0885(2) |
|  | -7.32980(18) | 0.14827(18) |  | -54.45705(15) | 0.1318(2) |
|  | -7.3285(3) | 0.1496(3) |  | -54.45691(16) | 0.1320(2) |
|  | -7.32391(19) | 0.15416(19) |  | -54.45680(15) | 0.1321(2) |
|  | -7.3150(4) | 0.1631(4) |  | -54.20615(17) | 0.3827(2) |
| Be | -14.66733(5) | - | O | -75.0661(2) | - |
|  | -14.56718(4) | 0.10015(6) |  | -75.0660(2) | 0.0001(3) |
|  | -14.56712(4) | 0.10021(6) |  | -75.0659(2) | 0.0001(3) |
|  | -14.56701(4) | 0.10031(6) |  | -74.9940(2) | 0.0721(3) |
|  | -14.47336(5) | 0.19397(7) |  | -74.9939(2) | 0.0722(3) |
|  | -14.47327(5) | 0.19405(7) |  | -74.9939(2) | 0.0722(3) |
|  | -14.47322(5) | 0.19411(7) |  | -74.9935(2) | 0.0725(3) |
|  | -14.42975(5) | 0.23758(7) |  | -74.9924(2) | 0.0737(3) |
|  | -14.41709(5) | 0.25024(7) |  | -74.9127(2) | 0.1533(3) |
|  | -14.40813(6) | 0.25920(8) |  | -74.7273(3) | 0.3388(3) |
| B | -24.65374(7) | - | F | -99.7296(3) | - |
|  | -24.65372(9) | 0.00002(11) |  | -99.7281(3) | 0.0016(4) |
|  | -24.65362(7) | 0.00012(9) |  | -99.7277(3) | 0.0019(4) |
|  | -24.52204(6) | 0.13171(9) |  | -99.2631(3) | 0.4665(4) |
|  | -24.52203(7) | 0.13171(9) |  | -99.2630(3) | 0.4667(4) |
|  | -24.52197(6) | 0.13177(9) |  | -99.2628(3) | 0.4668(4) |
|  | -24.47116(14) | 0.18258(15) |  | -99.2531(3) | 0.4765(4) |
|  | -24.43550(8) | 0.21825(11) |  | -99.2530(3) | 0.4767(4) |
|  | -24.43539(8) | 0.21835(11) |  | -99.2526(3) | 0.4770(4) |
|  | -24.43508(9) | 0.21866(11) |  | -99.1956(3) | 0.5341(4) |
| C | -37.84474(9) | - | Ne | -128.9337(3) | - |
|  | -37.84462(9) | 0.00012(13) |  | -128.3188(3) | 0.6149(5) |
|  | -37.84448(9) | 0.00026(13) |  | -128.3186(3) | 0.6151(5) |
|  | -37.79831(10) | 0.04643(13) |  | -128.3186(4) | 0.6151(5) |
|  | -37.79826(10) | 0.04647(14) |  | -128.3185(3) | 0.6152(5) |
|  | -37.79820(10) | 0.04654(14) |  | -128.3183(3) | 0.6154(5) |
|  | -37.79799(9) | 0.04674(13) |  | -128.3183(3) | 0.6154(5) |
|  | -37.79790(10) | 0.04684(13) |  | -128.2511(4) | 0.6826(5) |
|  | -37.74588(10) | 0.09885(14) |  | -128.2488(3) | 0.6849(5) |
|  | -37.69152(9) | 0.15322(13) |  | -128.2479(3) | 0.6858(5) |

*Table 11.* Absolute energies of the carbon dimer upon dissociation (Fig. 5) obtained from a jointly trained Excited Pfaffian. We report raw energies as tracked by our generalized wave function, before averaging over degenerate states and correcting for state swaps. Energies are in Hartree, bond lengths in Angstrom.

| Bond length | $X^1\Sigma_g^+$ | $A^1\Pi_{u+}$ | $A^1\Pi_{u-}$ | $B'^1\Sigma_g^+$ | $B^1\Delta_g^{(2)}$ | $B^1\Delta_g^{(1)}$ |
|---|---|---|---|---|---|---|
| 0.9952 | -75.77445(12) | -75.66149(12) | -75.65982(12) | -75.59821(14) | -75.57830(12) | -75.57735(12) |
| 1.1204 | -75.89580(11) | -75.82222(11) | -75.82083(11) | -75.76439(11) | -75.76767(11) | -75.76991(11) |
| 1.1823 | -75.91752(10) | -75.85855(10) | -75.85629(10) | -75.80813(11) | -75.81718(10) | -75.82005(10) |
| 1.2431 | -75.92326(10) | -75.87808(10) | -75.87552(10) | -75.83539(10) | -75.84666(10) | -75.84959(10) |
| 1.3064 | -75.91801(10) | -75.88474(10) | -75.88113(10) | -75.84960(10) | -75.86225(10) | -75.86555(10) |
| 1.3685 | -75.90679(10) | -75.88355(10) | -75.88037(10) | -75.85471(10) | -75.86803(10) | -75.87119(10) |
| 1.4933 | -75.87089(10) | -75.86346(10) | -75.86208(10) | -75.84603(10) | -75.86028(10) | -75.86090(10) |
| 1.6165 | -75.83451(10) | -75.83331(10) | -75.83790(10) | -75.82297(10) | -75.84134(10) | -75.83634(10) |
| 1.7430 | -75.79021(11) | -75.80243(10) | -75.81338(10) | -75.80548(10) | -75.81516(10) | -75.80486(11) |
| 1.8651 | -75.75787(11) | -75.77323(11) | -75.78945(11) | -75.78393(11) | -75.79125(11) | -75.77600(11) |

| Bond length | $c^3\Sigma_u^+$ | $a^3\Pi_{u-}$ | $a^3\Pi_{u+}$ | $d^3\Pi_{g-}$ | $d^3\Pi_{g+}$ | $b^3\Sigma_g^-$ |
|---|---|---|---|---|---|---|
| 0.9952 | -75.75867(11) | -75.70036(11) | -75.69934(11) | -75.65572(12) | -75.65556(12) | -75.61480(11) |
| 1.1204 | -75.86615(11) | -75.85982(10) | -75.86010(10) | -75.79421(11) | -75.79491(11) | -75.80112(10) |
| 1.1823 | -75.88022(10) | -75.89555(10) | -75.89619(10) | -75.82135(11) | -75.82175(11) | -75.84883(10) |
| 1.2431 | -75.87916(10) | -75.91446(10) | -75.91454(10) | -75.83181(10) | -75.83160(11) | -75.87619(10) |
| 1.3064 | -75.86714(10) | -75.92032(10) | -75.91953(10) | -75.83021(10) | -75.83006(10) | -75.89065(10) |
| 1.3685 | -75.84896(11) | -75.91663(9) | -75.91721(10) | -75.82018(10) | -75.82144(11) | -75.89423(10) |
| 1.4933 | -75.80489(11) | -75.89452(9) | -75.89491(10) | -75.79243(11) | -75.79292(11) | -75.88361(9) |
| 1.6165 | -75.76506(13) | -75.86480(10) | -75.86443(10) | -75.76466(12) | -75.76485(12) | -75.86101(10) |
| 1.7430 | -75.73960(14) | -75.83002(10) | -75.83058(10) | -75.74462(13) | -75.74419(14) | -75.83308(10) |
| 1.8651 | -75.73037(14) | -75.79987(11) | -75.79916(11) | -75.72998(14) | -75.73083(15) | -75.80611(11) |

*Table 12.* Absolute energies of the two lowest singlet states of ethylene (Fig. 7) under torsion and pyramidalization. A single Excited Pfaffian is trained jointly across both PES. Energies are in Hartree, angles in degrees.

*(a)* Ethylene PES under torsion $\tau$

| $\tau$ (degrees) | State 0 | State 1 |
|---|---|---|
| 0.0 | -78.58667(13) | -78.31829(14) |
| 15.0 | -78.58190(12) | -78.31864(14) |
| 30.0 | -78.33172(14) | -78.56958(12) |
| 45.0 | -78.35356(14) | -78.54934(12) |
| 60.0 | -78.36872(14) | -78.52137(12) |
| 70.0 | -78.37631(14) | -78.50073(12) |
| 80.0 | -78.37996(14) | -78.48113(12) |
| 85.0 | -78.38131(14) | -78.47387(12) |
| 90.0 | -78.38162(14) | -78.47169(12) |

*(b)* Ethylene PES under pyramidalization $\phi$

| $\phi$ (degrees) | State 0 | State 1 |
|---|---|---|
| 0.0 | -78.38574(14) | -78.47418(12) |
| 20.0 | -78.38798(13) | -78.47309(12) |
| 40.0 | -78.39538(12) | -78.46763(12) |
| 60.0 | -78.40160(12) | -78.45390(12) |
| 70.0 | -78.40232(12) | -78.44202(12) |
| 80.0 | -78.40123(12) | -78.42718(12) |
| 90.0 | -78.39607(12) | -78.40817(12) |
| 95.0 | -78.39202(12) | -78.39694(12) |
| 97.5 | -78.39010(12) | -78.39132(12) |
| 100.0 | -78.38749(12) | -78.38542(12) |
| 102.5 | -78.38470(12) | -78.37920(12) |
| 105.0 | -78.38183(12) | -78.37284(12) |
| 110.0 | -78.37459(12) | -78.36005(12) |
| 120.0 | -78.35782(13) | -78.33403(13) |

*Table 13.* Absolute and excitation energies of the 33 states of beryllium (Fig. 6). Reference energies from NIST $\Delta E_{\text{exp}}$ (Sansonetti, 2003) and errors $\Delta \Delta E_{\text{exp}}$ are also given. Energies are given in atomic units.

| State | Energy (Ha) | $\Delta E$ (Ha) | $\Delta E_{\text{exp}}$ (Ha) | $\Delta \Delta E_{\text{exp}}$ (Ha) |
|---|---|---|---|---|
| 0 | -14.66719(3) | - | - | - |
| 1 | -14.56714(6) | 0.10005(7) | 0.100149 | 0.000100 |
| 2 | -14.56707(3) | 0.10012(4) | 0.100149 | 0.000028 |
| 3 | -14.56706(3) | 0.10013(4) | 0.100149 | 0.000020 |
| 4 | -14.47333(3) | 0.19386(4) | 0.193942 | 0.000083 |
| 5 | -14.47327(3) | 0.19392(4) | 0.193942 | 0.000020 |
| 6 | -14.47300(3) | 0.19419(4) | 0.193942 | 0.000248 |
| 7 | -14.43005(3) | 0.23714(4) | 0.237298 | 0.000155 |
| 8 | -14.41846(3) | 0.24873(4) | 0.249128 | 0.000394 |
| 9 | -14.40822(3) | 0.25897(4) | 0.259175 | 0.000205 |
| 10 | -14.40822(3) | 0.25897(4) | 0.259175 | 0.000203 |
| 11 | -14.40819(3) | 0.25900(4) | 0.259175 | 0.000175 |
| 12 | -14.40818(3) | 0.25901(4) | 0.259175 | 0.000168 |
| 13 | -14.40815(3) | 0.25905(4) | 0.259175 | 0.000130 |
| 14 | -14.39924(3) | 0.26795(4) | 0.268403 | 0.000453 |
| 15 | -14.39914(3) | 0.26805(4) | 0.268403 | 0.000352 |
| 16 | -14.39907(4) | 0.26812(5) | 0.268403 | 0.000287 |
| 17 | -14.39549(3) | 0.27170(4) | 0.271995 | 0.000293 |
| 18 | -14.39547(2) | 0.27172(4) | 0.271995 | 0.000276 |
| 19 | -14.39546(3) | 0.27173(4) | 0.271995 | 0.000267 |
| 20 | -14.39330(3) | 0.27390(4) | 0.274234 | 0.000339 |
| 21 | -14.39329(3) | 0.27390(4) | 0.274234 | 0.000332 |
| 22 | -14.39323(4) | 0.27396(5) | 0.274234 | 0.000273 |
| 23 | -14.38462(3) | 0.28257(4) | 0.282738 | 0.000170 |
| 24 | -14.38462(3) | 0.28257(4) | 0.282738 | 0.000164 |
| 25 | -14.38461(3) | 0.28258(4) | 0.282738 | 0.000159 |
| 26 | -14.38457(3) | 0.28262(4) | 0.282738 | 0.000121 |
| 27 | -14.38457(3) | 0.28262(4) | 0.282738 | 0.000115 |
| 28 | -14.37392(3) | 0.29327(4) | 0.293557 | 0.000284 |
| 29 | -14.37390(3) | 0.29329(4) | 0.293557 | 0.000265 |
| 30 | -14.37388(3) | 0.29331(4) | 0.293557 | 0.000244 |
| 31 | -14.37382(3) | 0.29337(4) | 0.293557 | 0.000183 |
| 32 | -14.37377(4) | 0.29343(5) | 0.293557 | 0.000132 |

