# OpenReview forum: "Excited Pfaffians: Generalized Neural Wave Functions Across Structure and State"
_ICML.cc/2026/Conference — ICML 2026 spotlight_

### Official Review · Reviewer_yVhX · 2026-03-03

**Soundness:** 3
**Presentation:** 3
**Significance:** 4
**Originality:** 3
**Overall Recommendation:** 5
**Confidence:** 2

**Summary:**

The focus of the paper is on the problem of obtaining the eigenstates - particular the non-minimal eigenstates a.k.a. the excited states - of quantum systems by accelerating a standard tool, namely Variational Monte Carlo. The extension of the Neural Pfaffian approach to excited states is key to the proposed approach, as is the use of samples collected from all states (multi-state importance sampling).

**Compliance With Llm Reviewing Policy:**

Affirmed.

**Key Questions For Authors:**

Should I understand your sampling approach to be related to multiple importance sampling (e.g. https://lisyarus.github.io/blog/posts/multiple-importance-sampling.html)?

What MCMC kernel are you using? If this is a continuous state space, presumably you should be using a gradient based method for sampling to avoid extremely high sampling costs. My sense is that this is fairly uncommon in this field, and I'd be curious to understand why.

**Limitations:**

There was a reasonable discussion of limitations.

**Strengths And Weaknesses:**

In terms of significance, the paper seems quite impressive. The proposed method appears to be much faster than its competitors. The results also show nice applications that the approach enables, for obtaining the excited states of Beryllium, and ground states through crossings. This is a hard, well-studied problem, where improvements are impactful.

In terms of presentation, I'm not sure what the expectations in terms of accessibility are for general ML readers, but certainly concepts like fermions (or indeed any quantum physics) will be totally inaccessible if they are not phrased entirely in the language of linear algebra. And even for someone familiar with the language of quantum mechanics, the introduction to Hartree-Fock (which is key to understanding the paper) went very fast. Whether papers in this track should attempt to be legible to ML readers is a question for the area chair and others. It's certainly possible to write papers like this entirely in a language familiar to the general ICML audience, but maybe not easy.

I am not well-equipped to comment on the soundness of the method. As to the originality, there seem to be several novel contributions, although broadly the contribution seems to be the combination of known methods for NN amortization and sampling together.

---

> ### Author Rebuttal · Authors · 2026-03-30
>
> We thank the reviewer for the thoughtful evaluation. All experiments have been rerun for 200k steps, following prior works (Pfau et al. 2024, Szabó et al. 2024), to ensure fully converged energies. In the [supplementary PDF](https://figshare.com/s/98cae565c03eba8df308), we provide updated results for all systems (atoms, molecules, Be, C$_2$ PES, ethylene PES), the effective sample size across all systems, direct wall-clock runtime comparisons against NES, and per-step timing tables.
>
> **Presentation**: We have done our best to make the paper accessible, but acknowledge that bridging quantum chemistry and ML is challenging. For added clarity, we have made the fermionic antisymmetry requirement more explicit:
> > As electrons are fermions, the wave function must be antisymmetric under permutation of same-spin electrons, i.e., $\Psi(\pi(r))=\mathrm{sign}(\pi)\Psi(r)$ for $\pi\in S_{N_\uparrow}\times S_{N_\downarrow}$.
>
> And have made the connection between the Hartree-Fock trial wave function and the variational optimization problem more apparent by introducing the HF coefficients as
> > [...] variational parameters $C \in\Bbb{R}^{N_o \times N_o}$. The parameters $C$ minimize $E[\psi_{\mathrm{HF}}]$ under the orthogonality constraints $\bra{\phi_p}{\phi_q}\rangle = \delta_{pq}$.
>
> We would appreciate hearing further specific parts the reviewer found most difficult to follow so we can target our revisions effectively.
>
> **Originality**: Before this work, computing N excited states required N times the cost. Our contribution is identifying why (the overlap estimator is the bottleneck) and resolving it through MSIS and an architecture that makes multi-state evaluation nearly free. On C$_2$ with 12 states we measure 50s/step for NES vs 10.8s/step for ours, a $4.6\times$ speedup. This enabled us to be the first to compute all 33 states of beryllium. Figure 6 and Tables 1–4 (supplementary PDF) provide direct wall-clock comparisons and sample-count summaries.
>
> **Soundness**: We validate our results against established benchmarks: on beryllium, all 33 states match NIST reference values to within 0.5 m$E_h$ (Figure 3), and on C$_2$, our PES agrees with the accurate semistochastic heat-bath CI calculations (Figure 2).
>
> **Q1 (Multiple importance sampling)**: Yes, MSIS is closely related to the *balance heuristic* from Monte Carlo rendering (Veach et al. 1995) and the "deterministic mixture" estimator in statistics (Owen et al. 2000). While the application domain differs, the underlying estimator structure is the same and both works show it is provably near-optimal for combining samples from multiple proposals. Concretely, we integrate $f(\mathbf{x}) = \Psi_0(\mathbf{x})\Psi_1(\mathbf{x})$ using samples from $p_s(\mathbf{x}) = |\Psi_s(\mathbf{x})|^2$, giving the estimator $f(\mathbf{x})/\sum w_s p_s(\mathbf{x})$ with uniform weights $w_s = 1/N_s$. The denominator is $\rho_\mathrm{mix}$ from Eq. 10. We have added these references to the related work:
> > We propose pooling samples from all states via importance sampling to stabilize overlap estimates without increasing the batch size, an approach closely related to the balance heuristic in Monte Carlo rendering (Veach et al. 1995, Owen et al. 2000).
>
> **Q2 (MCMC kernel)**: We use Metropolis-Hastings with a Gaussian proposal, the de facto standard in neural-network VMC. Schätzle et al. (2023, Fig. 4) compared Metropolis and Langevin samplers and found that while Langevin achieves lower autocorrelation times, the additional cost of computing gradients through the network makes the wall-clock time per decorrelated sample comparable for both methods. Since walkers run continuously throughout training and are thus always warm-started, the simpler Metropolis sampler is sufficient.
>
> **References**
>
> Owen et al. Safe and effective importance sampling. 2000.
>
> Schätzle et al. DeepQMC: an open-source software suite for variational optimization of deep-learning molecular wave functions. 2023.
>
> Veach et al. Optimally combining sampling techniques for Monte Carlo rendering. 1995.

---

> > ### Author Rebuttal · Reviewer_yVhX · 2026-04-02
> >
> > Thanks, this was helpful!

---

### Official Review · Reviewer_3DzZ · 2026-03-12

**Soundness:** 3
**Presentation:** 3
**Significance:** 3
**Originality:** 3
**Overall Recommendation:** 4
**Confidence:** 2

**Summary:**

​	This paper studies the scalability issue of variational Monte Carlo with neural wave functions when the number of target states increases. The paper proposes MSIS, a multi-state importance sampling method, to reduce computational cost. In addition, it improves computational efficiency by sharing neural network parameters across different states, while using only lightweight state selectors to distinguish between them. The method is validated on multiple systems. The results show that, while maintaining high accuracy, the proposed approach achieves better scalability than existing methods.

**Compliance With Llm Reviewing Policy:**

Affirmed.

**Final Justification:**

The authors has addressed my concerns, and i will keep my positive score

**Key Questions For Authors:**

1. Although Appendix G explains that the “naive extension” leads to OOM at large scale, could the authors provide a small-scale ablation comparing Excited Pfaffian with the naive extension directly?
1. The paper claims to achieve generalized neural wave functions, but the current results seem closer to demonstrating joint training capability rather than stable and consistent generalization, and state switching still occurs. Could the authors more clearly clarify the actual capability boundary of the current method?

**Limitations:**

yes

**Strengths And Weaknesses:**

Strengths

1. The paper focuses on the scalability bottleneck in multi-state modeling, and the problem setting is clear and well motivated.
1. The method improves scalability from two aspects simultaneously: sampling estimation and parameter sharing, resulting in a relatively complete overall design.
1. The experiments are fairly comprehensive and demonstrate both the accuracy and the scaling efficiency of the proposed method.


Weaknesses

1. The runtime comparisons in the paper are mainly based on estimation, rather than end-to-end timing measurements.
1. The paper lacks sufficient ablation on the model architecture itself. The ablation study is mainly limited to MSIS.

---

> ### Author Rebuttal · Authors · 2026-03-30
>
> We thank the reviewer for the constructive assessment. All experiments have been rerun for 200k steps, following prior works (Pfau et al. 2024, Szabó et al. 2024), to ensure fully converged energies. In the [supplementary PDF](https://figshare.com/s/98cae565c03eba8df308), we provide updated results for all systems (atoms, molecules, Be, C$_2$ PES, ethylene PES), the effective sample size across all systems, direct wall-clock runtime comparisons against NES, and per-step timing tables.
>
> **W1 (Runtime)**: We now provide direct wall-clock measurements on Be and C$_2$ (Figure 6). On C$_2$ with 12 states, we measure a $4.6\times$ per-step speedup over NES with SPRING, and $\sim 21\times$ when both use Adam. The choice of optimizer is independent of our main contribution (MSIS and Excited Pfaffians). We used SPRING because of KFAC's implementation complexity. As visible in Figure 6, the overhead of SPRING is outsized on small systems and decreases relative to the rest of the VMC optimization with system size. Tables 1–3 provide real-world runtime measurements per training step for atoms, molecules, and PES respectively. Table 4 consolidates sample counts.
>
> **W2 / Q1 (Architecture ablation)**: To fit the naive approach into memory, we reduced the number of Pfaffians/Determinants from 16 to 1.We plot the convergence of the total energy of the 33 states of Be in Figure 7 in the supplementary PDF. Due to its large tensors, the naive approach takes 48s per step, compared to our 8s per step. Unfortunately, we will not able to fully converge this model as it would take approx. 111 A100 compute days. For comparison, we also trained an additional Excited Pfaffian with 1 instead 16 det (7.2s/step) and find it to agree well in energy with the naive approach.
>
> **Q2 (Capability boundary)**: In our updated results, our method reliably resolves well-separated states: Be with 33 states (Figure 3), C$_2$ with 12 states across 10 geometries (Figure 2), and generalization across second-row atoms and molecules (Figure 1). We define generalized wave functions in Sec. 2 following Gao et al. (2023) and have added:
> > Note that the transfer to unseen structures has seen less success and is out of scope of this work (Scherbela et al. 2023).
>
> We observe two types of edge cases (discussed in the Limitations of Sec. 4). The first is fundamental to numerical solutions within the Born-Oppenheimer approximation. The second is a capacity issue with known mitigations.
>
> 1. *State discontinuities* occur where states change character discontinuously (avoided crossings, conical intersections), which smooth parameterization cannot capture. Schätzle et al. (2025), the only other work with state tracking, observes the same. We have expanded the paper. On ethylene:
>     > We observe a loss of state tracking in the torsion PES between 15° and 30°, where a Rydberg state crosses the valence excited state resulting in a change in the electronic character of the upper state, see App. J (Barbatti et al., 2004).
>
>     On C$_2$:
>     > Like Schätzle et al. (2025), we observe a single state swap between the structurally similar $X^1\Sigma_g^+$ and $B'^1\Sigma_g^+$ states near 1.6A. This reflects a narrow avoided crossing (von Neumann et al. 1929) at which the dominant electronic character exchanges over a small range of bond lengths.
>
>     And to the Limitations section:
>     > More generally, perfect state continuity is unattainable near conical intersections, where states become degenerate and their character changes discontinuously (Yarkony, 1996).
>
> 2. *Linear combinations of near-degenerate states* (H2O, H2CSi, C2H4) are a separate phenomenon also observed in single-structure baselines (Szabó et al. 2024). With extended training, these have improved significantly. We expect larger models to further resolve them. Remaining linear combinations could also be resolved at inference time via the NES framework by estimating $\mathbf{S}$ and $\mathbf{H}$ (def. as in Pfau et al. 2024) from $\rho_\mathrm{mix}$ samples and computing the eigenvalues of $\mathbf{S}^{-1}\mathbf{H}$, which we leave for future work.
>
> **References**
>
> Gao et al. Ab-initio potential energy surfaces by pairing GNNs with neural wave functions. 2022.
>
> Gao et al. Generalizing neural wave functions. 2023.
>
> Gerard et al. Transferable neural wavefunctions for solids. 2025.
>
> Pfau et al. Accurate computation of quantum excited states with neural networks. 2024.
>
> Scherbela et al. Variational Monte Carlo on a budget -- Fine-tuning pre-trained neural wavefunctions. 2023.
>
> Scherbela et al. Towards a transferable fermionic neural wavefunction for molecules. 2024.
>
> Yarkony, D. R. Diabolical conical intersections. 1996.

---

> > ### Author Rebuttal · Reviewer_3DzZ · 2026-04-03
> >
> > Thank you for the response, I will keep my positive score.

---

### Official Review · Reviewer_LywF · 2026-03-13

**Soundness:** 3
**Presentation:** 3
**Significance:** 2
**Originality:** 3
**Overall Recommendation:** 5
**Confidence:** 3

**Summary:**

This paper studies how to scale excited-state neural-network variational Monte Carlo across many states and across molecular structure. The paper proposes two main ideas: Multi-State Importance Sampling (MSIS), which pools samples from all states to estimate pairwise overlaps more efficiently, and Excited Pfaffians, a state-shared Pfaffian wave-function architecture in which most computation is shared across states and only lightweight selectors are state-specific. The paper also introduces spin-state snapping and a pretraining/alignment procedure for tracking states across geometries. Experiments evaluate runtime scaling, joint training on second-row atoms and molecules, many-state beryllium, carbon-dimer and ethylene potential-energy surfaces, an MSIS ablation, and a state-crossing example, with comparisons to recent excited-state neural VMC methods such as Pfau et al. (2024), Szabo et al. (2024), and Schatzle et al. (2025).

**Compliance With Llm Reviewing Policy:**

Affirmed.

**Final Justification:**

My main concern is resolved.

**Key Questions For Authors:**

1. Could the authors provide at least one direct, matched wall-clock comparison against NES or another strong excited-state neural VMC baseline on a common configuration? A direct comparison would materially increase my confidence in the practical efficiency claims.

2. Could the authors clarify the intended take-away from the ethylene experiments, especially the distinction between individual-slice training and the jointly trained setting? A clearer explanation here would help me judge how strongly the current results support multi-geometry state tracking.

3. Could the authors report robustness statistics for convergence and state tracking, such as how often runs remain unconverged after the default budget, how often wrong-spin snapping occurs, or how often state discontinuities are observed? This would improve my assessment of reliability.

4. Could the authors provide one consolidated summary table that separates measured step-time scaling, sample-count reductions, and estimated total compute savings? Right now these efficiency claims are spread across the main text and appendix, which makes them harder to evaluate precisely.

**Limitations:**

Yes

**Strengths And Weaknesses:**

**Soundness.** From a soundness perspective, the paper is technically serious and generally well motivated. MSIS directly targets the overlap-estimation bottleneck, and the Excited Pfaffian architecture is a principled way to share computation across states while retaining a structured antisymmetric wave-function form. The empirical section is also broader than a single proof of concept, covering runtime scaling, cross-compound training, many-state beryllium, potential-energy surfaces, an MSIS ablation, and a state-crossing example. My reservation is that some of the strongest efficiency claims are supported somewhat indirectly. In particular, the main scaling comparison is not a fully matched end-to-end runtime comparison to prior methods such as NES (Pfau et al., 2024), and some of the largest compute-savings numbers are estimate-based rather than entirely measured in a controlled setting. In addition, the appendix notes that some experiments were not fully converged under the default training budget and required additional training, which makes the practical picture somewhat less clean than the headline framing suggests.

**Presentation.** The paper is generally well written and the overall narrative is easy to follow. I appreciated that the method section separates the sampling contribution from the architectural one, and the figures make the intended benefits relatively clear. The appendix also contains useful implementation and compute details. My main presentation concern is that several important caveats are easier to find in the appendix than in the main text, including the role of estimate-based compute comparisons, the fact that some experiments required longer training, and the limitations of state continuity in joint multi-geometry settings. I would encourage the authors to surface these points more prominently in the main paper. I would also like a clearer explanation of the ethylene setup, since the paper distinguishes between individual-slice training and a jointly trained setting that shows a state discontinuity.

**Significance.** I do think the paper addresses an important problem. Excited-state neural VMC is expensive, overlap estimation is a real bottleneck, and methods that amortize across states and structures could matter for future work in scientific machine learning and computational chemistry. The results on carbon dimer and many-state beryllium are particularly interesting, and the cross-compound experiments are a meaningful step beyond single-system demonstrations. That said, I am only moderately convinced by the breadth of the demonstrated impact. The paper itself notes that lossless generalization across chemical compounds remains open, and it acknowledges that state continuity is not guaranteed and accuracy may degrade on distinct compounds, similar to concerns already discussed in related generalized-wave-function work such as Gao & Gunnemann (2024) and Gerard et al. (2025). For that reason, I view the paper as a promising direction rather than fully convincing evidence of a broadly reliable generalized excited-state model.

**Originality.** On originality, I am positive. This does not read as a minor variant of an existing excited-state neural VMC method. The novelty comes from a thoughtful combination of ideas: pooled importance sampling for overlap estimation, a Hartree-Fock-inspired state-shared Pfaffian parameterization, dynamic spin-state snapping, and a pretraining/alignment scheme for multi-structure excited states. The reasoning behind this combination is well articulated, and the paper clearly aims at a real bottleneck in the literature relative to prior excited-state approaches such as Pfau et al. (2024), Szabo et al. (2024), and transferable deep QMC work like Schatzle et al. (2025). My overall stance remains borderline negative not because the core idea lacks originality, but because the empirical evidence still feels somewhat narrower and more indirectly supportive than the broadest framing suggests.

---

> ### Author Rebuttal · Authors · 2026-03-30
>
> We thank the reviewer for the detailed evaluation. All experiments have been rerun for 200k steps with fully converged energies. In the [supplementary PDF](https://figshare.com/s/98cae565c03eba8df308), we provide updated results for all systems (atoms, molecules, Be, C$_2$ PES, ethylene PES), the effective sample size across all systems, direct wall-clock runtime comparisons against NES, and per-step timing tables.
>
> **Q1 (Direct wall-clock comparison)**: Figure 6 shows wall-clock comparisons against NES on Be and C$_2$. On C$_2$ with 12 states, we measure 50s/step for NES vs 10.8s/step for ours (including SPRING), a $4.6\times$ per-step speedup per structure. With Adam, this grows to $\sim 21\times$ (Figure 6, bottom). Our method additionally amortizes across geometries (training 10 structures costs the same as one), yielding $\sim 46\times$ and $\sim 208\times$ total on the C$_2$ PES, with SPRING and Adam respectively. We would like to highlight that our method is optimizer agnostic and we use Spring for its simpler implementation. At equal compute, NES would only run $\sim 4.4$k steps per structure, roughly 2% of its own 200k-step budget (Pfau et al. 2024), insufficient for full convergence.
>
> We empirically validate this by training NES for 4.5k steps on the first structure of C2. The results are shown in Table 8 of our supplementary PDF. We find none of the NES states to be well converged.
>
> **Q2 (Ethylene)**: We trained a single model jointly on both PES (Figure 4) with excellent agreement across both surfaces. The only exception is a state swap at 15°–30° torsion, where a Rydberg state crosses the valence excited state (Barbatti et al. 2004). In the camera-ready version, we drop the individual-slice experiment and update the discussion to:
> > We observe a loss of state tracking in the torsion PES between 15° and 30°, where a Rydberg state crosses the valence excited state resulting in a change in the electronic character of the upper state, see App. J (Barbatti et al., 2004).
>
> **Q3 (Robustness)**:
> - *Convergence*: We adopted a systematic protocol of 200k steps for all experiments, matching prior works. This way, we found no experiment underconverged and all energies are stable throughout the last thousands of iterations.
> - *Spin snapping*: We apply spin-state snapping only to Be and molecules, activating it after 100k steps. For ethylene and C$_2$, we enforce the spin state explicitly. For atoms, we found spin snapping to exhibit high variance beyond singlet/triplet. We observed no incorrect assignments with this protocol.
> - *State swaps*: We observe state swaps in two cases (ethylene and C$_2$), both with physical origins. Only Schätzle et al. (2025) also tracks states across geometries and observes the same swap. For ethylene see above, for C$_2$, we update our discussion to:
>     > Like Schätzle et al. (2025), we observe a single state swap between the structurally similar $X^1\Sigma_g^+$ and $B'^1\Sigma_g^+$ states near 1.6A, which share the same symmetry and lie close in energy. This state swap reflects a narrow avoided crossing (von Neumann et al. 1929) at which the electronic character exchanges over a small range of bond lengths.
>
> **Q4 (Consolidated table)**: Table 4 consolidates sample counts: 800M for all our experiments, compared to 16–64B (Pfau et al. 2024), 8–16B (Szabó et al. 2024), and 3.2B (Schätzle et al. 2025). Tables 1–3 provide real-world runtime measurements per training step for atoms, molecules, and PES respectively.
>
> **Significance**: We agree that amortization across states and structures is important. Previous works scale at least linearly in the number of states, while ours achieves sublinear scaling with a measured $4.6\times$ speedup ($20\times$ with Adam). This enabled the first computation of all 33 states of beryllium and 12-state PES at a fraction of prior cost. As the reviewer notes, lossless generalization across compounds remains open. The cross-compound results (Figure 1) are an additional capability, not the sole justification.
>
> **References**
>
> Barbatti et al. Photochemistry of ethylene: A multireference configuration interaction investigation of the excited-state energy surfaces. 2004.
>
> Pfau et al. Accurate computation of quantum excited states with neural networks. 2024.
>
> Gao et al. Neural Pfaffians: Solving many many-electron Schrödinger equations. 2024.
>
> Schätzle et al. Ab-initio simulation of excited-state potential energy surfaces with transferable deep quantum Monte Carlo. 2025.
>
> Szabó et al. An improved penalty-based excited-state variational Monte Carlo approach with deep-learning ansatzes. 2024.
>
> von Neumann et al. Über das Verhalten von Eigenwerten bei adiabatischen Prozessen. 1929.

---

> > ### Author Rebuttal · Reviewer_LywF · 2026-04-03
> >
> > Thanks very much for the reply. My questions are resolved, and I will adjust my scores.

---

### Official Review · Reviewer_e6C7 · 2026-03-17

**Soundness:** 3
**Presentation:** 4
**Significance:** 3
**Originality:** 3
**Overall Recommendation:** 5
**Confidence:** 4

**Summary:**

This paper is about developing neural quantum state approaches for evaluating excited states.   To do this, they generate a number of trial neural quantum states which individually represent the n'th excited state.  By using a variational principle (eqn 7 in their work) they can simultaneously ensure that each excited state is a variational upper bound for their individual eigenstate. This has been done before.  There novel contributions are modifications that make it work better. The trial states are then represented as a pfaffian with orbitals taken from the neural network output of a configuration (fixed for all the states) and the skew-symmetric matrix taken from a state-dependent neural network.    To evaluate their variational principle, they need to compute the energy of each individual state which they do in the standard way.  To compute the the overlap they use samples from all of their calculations and then use reweighting to explicitly compute the overlap.  Additional tricks include ensuring that their state are not linear combination of S^2 eigenstates and a different way to do pretraining.   Additionally they train their wave-function so it is parameterized by the nuclei location so they can do potential energy surfaces easily.  They show a significant amount of numerics and demonstrate that they get reasonable results at small cost.

**Compliance With Llm Reviewing Policy:**

Affirmed.

**Final Justification:**

I scored this highly originally and the responses to my questions further supported that conclusion.

**Key Questions For Authors:**

1. What is the number of effective points you have when you compute the overlap with reweighting.  I would expect in the large system limit essentially all points that were not sampled with the wave-functions you are evaluating the overlap with respect to would reweight to zero and the effective number of points would go to 2/N_b which would be problematic with respect to the scaling. Could you measure this particularly as the system gets large and make a convincing case that this is not what is happening?

2.  Why doesn't the energy term of the loss function get significantly worse if you're using a number of configurations that goes to zero as you increase the number of excited state configurations?

**Limitations:**

yes

**Strengths And Weaknesses:**

The "weakest" part of this work is that the high-level approach of using NQS in this way to compute excited states has already been done and so the high-level conceptual novelty is limited. That said, the technical improvements they have made to the approach are relatively novel and make a pretty big difference with respect to the scaling (and to some extent the quality).  The biggest conceptual improvement is coming from using the samples from all the configurations to compute the overlap so that they can keep the total number of samples constant.  This idea is actually pretty natural but I'm relatively surprised that it worked.  In fact, my primary questions below are asking for a better understanding of why in the limit of a large number of excitations and/or large system sizes this doesn't fall back to the old process.  They also had a nice conceptual approach to getting enough variational space while not reusing too many parameters.  Finally they've done an impressive job demonstrating that numerically they are getting better/faster results then previously.      The only other weakness here is I'm not really convinced this approach will survive as the systems and number of excitations get larger.

---

> ### Author Rebuttal · Authors · 2026-03-30
>
> We thank the reviewer for the insightful questions. All experiments have been rerun for 200k steps, following prior works (Pfau et al. 2024, Szabó et al. 2024), to ensure fully converged energies. In the [supplementary PDF](https://figshare.com/s/98cae565c03eba8df308), we provide updated results for all systems (atoms, molecules, Be, C$_2$ PES, ethylene PES), the effective sample size across all systems, direct wall-clock runtime comparisons against NES, and per-step timing tables.
>
> **Q1**: Even if the ESS were low as the reviewer hypothesizes, this would imply that the uninformative states are spatially non-overlapping with the target pair. We mention this complementary mechanism in l. 383 (right block) to 398 (left block) in the manuscript:
> > For pairs with high similarity, samples from one state remain informative for the other, effectively increasing the sample size for overlap estimation. For distant pairs, the overlap estimand $\Psi_s\Psi_t/\rho_\mathrm{mix}$ becomes small in magnitude, causing the variance to vanish.
>
> For clarification we update this to:
> > For pairs with high similarity, samples from one state remain informative for the other, effectively increasing the sample size for overlap estimation. For distant pairs, our estimator's variance vanishes, see App. D.
>
> As we show in App. D, for wave functions with zero Bhattacharyya similarity, our estimator's variance is zero, while the single-state estimator retains unit variance. Even in the worst case, MSIS provides a better estimator.
>
> Empirically, the normalized effective sample size (ESS) is approximately constant. We measured the normalized ESS ($\times N_s / N_b$) of $\hat{O}_{ij}$ in Eq. (45) for all systems ([Figure 5](https://figshare.com/s/98cae565c03eba8df308)), where 1 corresponds to the reviewer's worst case ($\frac{N_b}{N_s}$ effective points). ESS remains well above 1: $\sim 3.7$–$4.7$ for atoms (3–10 valence electrons) and $\sim 1.8$–$2.8$ for molecules (up to 16 valence electrons), with no systematic decay with system size (Figure 5b,c). We leave the systematic scaling to larger systems for future work.
>
> **Q2**: The energy term is insensitive to the number of samples per state because the local energy estimator has inherently low variance (zero-variance principle), so even few samples yield accurate gradients. The overlap is the bottleneck as $N_s$ grows.
>
> The local energy $E_\text{loc} = \hat{H}\psi_s/\psi_s$ has the same wave function in numerator and denominator, so for good wave functions the variance is small (for exact eigenstates it is zero). The overlap estimator $\psi_t/\psi_s$ involves two different wave functions whose nodes do not coincide, causing divergent ratios and heavy-tailed outliers. This is what MSIS addresses.
>
> Empirically, training with constant total batch size across many structures does not degrade energy accuracy (Foster et al. 2025, Gao et al. 2022). We confirm this in Sec. 5 with only 32 samples per wave function on the C$_2$ PES. We add clarification to this in our background section to VMC:
> > Since $E_\mathrm{loc}$'s variance vanishes for exact eigenstates (zero-variance principle), splitting the sample budget across structures does not degrade energy accuracy (Gao & Günnemann, 2022).
>
>
> **References**
>
> Foster et al. An ab initio foundation model of wavefunctions that accurately describes chemical bond breaking. 2025.
>
> Gao et al. Ab-initio potential energy surfaces by pairing GNNs with neural wave functions. 2022.

---

> > ### Author Rebuttal · Reviewer_e6C7 · 2026-04-04
> >
> > Thanks for the clarifications.  This has resolved my questions.  I will keep my positive score.

---

### Decision · Program_Chairs · 2026-04-30

**Decision:**

Accept (spotlight)

**Comment:**

This paper proposes two contributions for scaling excited-state neural network VMC: multi state importance sampling and excited Pfaffians. The new method achieves near constant scaling and allows first neural network-based computation of 33 beryllium excited states and 12 state C2 potential energy surface.

All four reviewers recommend acceptance. Concerns were centered on indirect runtime comparisons, limited architecture ablations, and unclear generalization boundaries. The rebuttal comprehensively addressed these by providing direct wall-clock measurements, architecture ablations, extended training with full convergence on 200k-step results, and clarifications of limitations. All reviewers acknowledged their concerns as fully resolved.